# Task-sharing for non-communicable disease prevention and control in low- and middle-income countries in the context of health worker shortages: A systematic review

Azeb Gebresilassie Tesema[1,2]*, Sikhumbuzo A. Mabunda[1,2,3], Kanika Chaudhri[2], Anthony Sunjaya[2], Samuel Thio[1], Kenneth Yakubu[2], Ragavi Jeyakumar[1], Myron Godinho[4], Renu John[5], Mai Eltigany[6], Martyna Hogendorf[6‡], Rohina Joshi[1,2,4‡]

**1** School of Population Health, UNSW, Sydney, Australia, **2** The George Institute for Global Health, UNSW, Sydney, Australia, **3** Department of Public Health, Walter Sisulu University, Mthatha, South Africa, **4** Westmead Applied Research Centre, University of Sydney, Australia, **5** The George Institute for Global Health, UNSW, Delhi, India, **6** World Health Organization, Geneva, Switzerland.

‡ These authors are joint senior authors on this work.
* a.tesema@unsw.edu.au

## Abstract

Health workers are pivotal for non-communicable disease (NCD) service delivery, yet often are unavailable in low- and middle-income countries (LMICs). There is limited evidence on what NCD-related tasks non-physician health workers (NPHWs) can perform and their effectiveness. This study aims to understand how task-sharing is used to improve NCD prevention and control in LMICs. We also explored barriers, facilitators, and unexpected consequences of task-sharing. Databases searched in two phases and included MEDLINE, EMBASE, CENTRAL, CINAHL, Cochrane, and clinical trial registries, and references of included studies from inception until 31st July 2024. We included randomised control trials (RCTs), cluster RCTs, and associated process evaluation and cost effectiveness studies. The risk of bias was assessed using the Cochrane Risk of Bias Tool v2. PROSPERO: CRD42022315701. The study found 5527 citations, 427 full texts were screened and 149 studies (total population sample>432567) from 31 countries were included. Most studies were on tasks shared with nurses (n=83) and community health workers (n=65). Most studies focussed on cardiovascular disease (n=47), mental health (n=48), diabetes (n=27), cancer (n=20), and respiratory diseases (n=10). Seventeen studies included two or more conditions. Eighty-one percent (n=120) of studies reported at least one positive primary outcome, while 19 studies reported neutral results, one reported a negative result, eight (5.4%) reported mixed positive and neutral results, and one reported neutral and negative findings. Economic analyses indicated that task-sharing reduced total healthcare costs. Task-sharing is an effective intervention for NCDs in LMICs. It is essential to enhance the competencies and training of NPHWs, provide resources to augment their capabilities, and formalise their role in the health system and community. Optimising task-sharing for NCDs requires a holistic approach that strengthens health systems while supporting NPHWs in effectively addressing the diverse needs of their communities.

**Data availability statement:** Data collected for the study is provided within the manuscript. Additional related documents are available in the supplementary section.

**Funding:** This review was funded by the World Health Organization (WHO). ME and MH received salaries from WHO. The funders had no role in study design, data collection and analysis, decision to publish, or preparation of the manuscript.

**Competing interests:** The authors have declared that no competing interests exist.

**Registration:** PROSPERO CRD42022315701.

## Background

Low and middle-income countries (LMICs, as defined by the World Bank) have a rising prevalence of non-communicable diseases (NCDs)[1,2]. They have a proportionately younger population, and yet their age-standardised mortality rate for cardiovascular diseases (CVD) is greater than that of higher-income nations[1]. People living with NCDs rely on health systems to deliver a continuum of appropriate, affordable, and high-quality services for preventing, treating, and rehabilitating NCDs. This global trend necessitates that health services transition towards models of care that are patient centred, accessible to communities, and which improve health outcomes[3]. The World Health Organization (WHO) has committed to strengthen and orient health systems to address NCDs through integrated people-centred primary health care, towards achieving universal health coverage (UHC)[4]. A set of cost-effective interventions are further recommended for wide implementation to assist countries in reaching global targets for NCDs[5].

Health workers are pivotal for NCD service delivery, yet often remain the limiting factor in health systems due to shortages or lack of training[6,7]. In order to meet UHC targets, the world needs more than 43 million additional health workers. Estimates suggest that per 10,000 population, countries need at least 20.7 physicians, 70.6 nurses and midwives, 8.2 dental personnel, and 9.4 pharmaceutical workers to achieve an effective coverage index score of 80 out of 100 [8]. The most acute health workforce shortages are experienced in LMICs, particularly in sub-Saharan Africa, South Asia, North Africa and the Middle East[8]. LMICs are faced with critical decisions on how to "shape" the health workforce to be fit-for-purpose, ensuring that future and current health workers have the required competencies, supervision, resources, and motivation to deliver quality care. An emerging approach for addressing this workforce need is 'task sharing,' which comprises the redistribution of health care tasks within workforces and communities[9]. According to Orkin et al., this occurs "when tasks are completed collaboratively between providers with different levels of training"[9].

The current evidence on which occupational groups can perform which tasks is limited[10]. Occupational groups with a shorter duration of pre-service education (i.e. community health workers (CHWs), non-physician clinicians, etc.) have seen a continual expansion of their tasks, based on population needs, yet their roles sometimes lack clear definition. Evidence indicates that non-physician health workers (NPHWs) (e.g. community health workers, nurses) can deliver various aspects of healthcare traditionally considered to require a physician. Although, this comes with inadequate regulatory protection, supervision, guidance, training, etc. [11]. A 2019 overview of systematic reviews analysed the barriers and facilitators to the delivery of care for NCDs by NPHWs in LMICs and provided high-level recommendations for health systems considering the adoption of task-sharing approaches [11]. However, being an overview of reviews, this study did not inspect individual interventions to identify their models of care or understand how tools and mechanisms were used to enable task-sharing. Therefore, this systematic review aims to understand the effectiveness of task-sharing and how it is used to improve NCD prevention and control in LMICs. We also explore the barriers, facilitators, and unexpected consequences of task-sharing.

## Methods

This systematic review assessed the task sharing for NCDs in LMICs by NPHWs. PROSPERO CRD42022315701.https://www.crd.york.ac.uk/prospero/display_record.php?ID=CRD42022315701

## Search strategy and selection criteria

We search MEDLINE, EMBASE, Cochrane, CENTRAL (Cochrane Central Register of Controlled Trials) and CINAHL in two phases; initially from the beginning of each database until 4th March 2022[12] and then updated the search from 1st March 2022 to 31st July 2024. Further studies were obtained from scanning reference lists of relevant studies and citation searching of key papers identified for inclusion. We searched references obtained from Cochrane Database of Systematic Reviews and search trial databases such as Clinicaltrials.gov for relevant studies. A search strategy was developed with the support of a medical librarian. We used Covidence to conduct the review[13]. Search terms are included in S1 Appendix. The following outcomes were assessed:

1. Which interventions related to prevention and control of NCDs (including prevention, promotion, management, rehabilitation, and palliation) are delivered by non-physician health workers? This included patient related outcomes (e.g., blood pressure control for hypertension related studies) indicating effectiveness of intervention delivery. We also reviewed system-related outcomes (e.g., NPHW workload) and unintended consequences of task-sharing (e.g. any harm caused).

2. Enablers and barriers for task-sharing for NCD prevention and control

Task-sharing refers to the redistribution of healthcare tasks across providers with varying levels of training to address workforce shortages. This can involve expanding the roles of existing health workers, such as nurses or CHWs or incorporating additional resources like volunteers or faith healers. The approach often utilizes a multidisciplinary team, which may include CHWs, nurses, and, in some cases, physicians [9].

Inclusion criteria comprised health facilities and communities in LMICs. Interventions involved NPHWs delivering prevention, screening, management, referral, rehabilitation, palliation for NCDs (such as diabetes, CVD, respiratory diseases, mental health disorders, cancer). Studies were included if physicians were involved with NPHWs as part of a multidisciplinary team.

This systematic review included randomised control trials (RCTs), cluster RCTs, and their associated process evaluation and cost-effectiveness studies. We included studies published in English, French and Spanish. Articles were excluded if they were not a peer reviewed article, not a report based on empirical research, pilot studies, not reported in English, Spanish or French, and research conducted on non-human subjects. Additionally, studies with fewer than 50 participants were excluded based on the sample size criterion. In both phases of search, two researchers independently reviewed and selected studies and articles against the inclusion criteria. Discrepancies between the reviewers were resolved by consultation with the team led by a third reviewer. In the case of duplicate reports, the paper with the most information was included.

## Data management and analysis

The shortlisted articles were exported to Endnote X9 (Thomson Reuters, NY, USA) for storage of study records, abstracts, and full text articles [14]. Data was stored on a password protected server-based platform that was accessible by the reviewers. Covidence, a systematic review platform, was used to streamline the process of reviewing articles. Data were collected using a standardised data extraction form. The form was piloted and optimised by two reviewers using a subset of three randomly selected studies that satisfied the eligibility criteria. Information outlined in the standardised data extraction form was collected by two reviewers independently. Data was cleaned and analysed using narrative synthesis. This was supplemented with tables and figures where appropriate. We used the PRISMA guidelines to optimize the quality of reporting.

### Risk of bias assessment

The risk of bias was assessed by two independent reviewers using the Cochrane Risk of Bias Tool v2[15, 16]. The assessment was performed at study level and focused on selection, performance, detection, attrition, and reporting bias. We did not exclude studies with a high risk of bias as we wanted to include all contexts. Furthermore, it is known that adhering to all critical aspects of study design is not always feasible in the health system setting, making some trials more vulnerable to bias[17].

### Role of funding source

The funder of the study had no role in study design, data collection, data analysis or data interpretation.

## Results

### Search and study selection

The search retrieved 4858 potentially relevant studies in the first phase and 669 articles in the second phase, totalling 5527 studies. In the first phase, 1372 duplicates were removed, and five duplicates were removed in the second phase. After an initial screening of title and abstract of 4150 articles (3486 citations (phase 1) and 664 (phase 2)), 427 (399 (phase 1) plus 28 (phase 2)) full text articles were assessed of which 278 (261 (phase 1) plus 17 (phase 2)) did not meet the eligibility criteria. A total of 149 (138 (phase 1) and 11 (phase 2)) studies were included in the final review (Fig 1).

### Summary of included studies

One-hundred and forty-nine (149) RCTs representing at least a total of 432,567 patients were included in this review. This is because one study was a cluster randomised controlled trial with the intervention implemented and assessed at household level (29000 households). This current study did not impute the average number of individuals in each household but instead used a ratio of 1:1 for each household and individuals to get the absolute minimum sample size. The smallest study included 50 participants[18] and the two largest studies included 151,538 participants from the same cohort, which we counted once[19, 20]. The third largest studies [21, 22] included 33,995 participants from the same cohort, which we also counted once. Table 1 summarises the characteristics of included studies. Trials were published between 2001[23] and 2023[24–28] in peer reviewed journals. On average, about 10 studies were published each year since 2014-2022, with the highest being 20 published in 2020. Almost two-thirds of the studies were conducted in Asia (64%, 96/149), with 21% (31/149) from Africa, and the rest from Europe (5%, 7/149), South America (9%, 14/149), and 1% (1/149) in each of Oceania and North America. One multi-centre study was done in both Asia and South America[29].

The studies originated from a total of 31 countries and there were five multi-country studies[29–33](Fig 2). Most studies (99%, 147/149) were in English and, one each in Spanish and in French. Fifty percent (75/149) of studies were conducted in urban compared to 26% (38/149) in rural areas, 17 studies (11%) were conducted in both urban and rural areas. The remaining, 13% (19/149) did not specify where they were conducted. Most studies (54%, 80/149) were conducted in the community, and 44% (66/149) were conducted in health centres or hospitals, with five studies in a combination of these settings[29–31,33,34]. Thirteen studies had a published process evaluation, and 11 studies conducted a cost-effectiveness evaluation. Workforce related outcomes (e.g. workload, frustration, satisfaction) were reported by studies that included process evaluations.

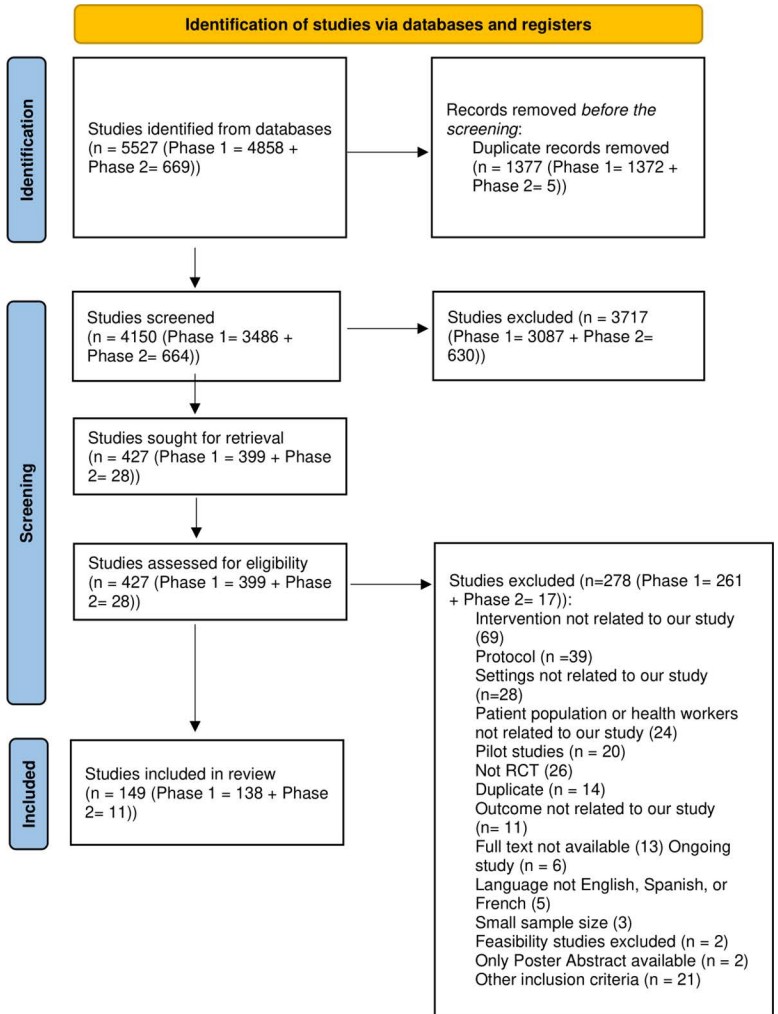

**Fig 1.  PRISMA Flow chart of study retrieval and selection.**

## Workforce involvement in task-sharing interventions

The workforce varied across different studies and contexts with 44% (65/149) interventions employing CHWs, and 56% (83/149) of studies using nurses (Table 2). Four studies included only dietitians or nutritionists [35–37] and three study lay health workers. Interventions included a range of services relating to prevention and health promotion (10%), screening (3%), management (36%), rehabilitation (3%) and palliation (1%). Majority (47%) involved a combination of activities including prevention, screening, management, referral and rehabilitation.

By comparison, CHWs were mostly involved in health education, screening and referrals. CHWs also conducted management in few studies. Some studies had multidisciplinary teams including the primary healthcare (PHC) team of CHWs, nurses, PHC doctor and a specialist or team of specialists[38–44]. Most studies reported providing training or employed a trained health workforce (69%, 103/149) and 39% (59/149) of studies reported having a supervisory structure for the workforce. None of the studies indicated evidence of harm of task-sharing NCD related interventions.

Table 1. Characteristics of articles.

| Author, Year | Country | Setting | Disease or condition | Participant number | Workforce included | Intervention impact* |
|---|---|---|---|---|---|---|
| Adewuya AO, et al.[38] 2019 | Nigeria | Rural; Urban | Mental Health | 907 | Nurse; Midwife; CHW; Pharmacy technicians | Positive |
| Adeyemo A, et al.[87] 2013 | Nigeria | Rural; Urban | CVD | 668 | Nurse + Doctor | Neutral |
| Ali M, et al.[39] 2020 | India | Urban | NCD combination | 404 | CHW + Care coordinator, psychiatrist, diabetologist | Positive |
| Al Ksir K, et al.[100] 2022 | Tunisia | Urban | Diabetes | 66 | Nurse | Positive |
| Arjunan P, et al. [101] 2021 | India | Rural; Urban | CVD | 200 | Nurse | Positive |
| Arrossi S, et al.[102] 2015 | Argentina | Rural; Urban | Cancer | 6013 | CHW | Positive |
| Azami G, et al. [103] 2018 | Iran | Urban | Diabetes | 142 | Nurse | Positive |
| Bass J, et al. [104], 2016 | Iran | Rural | Mental Health | 209 | CHW | Positive |
| Beratarrechea A, et al. [49], 2019 | Argentina | Urban | CVD | 755 | CHW | Positive |
| Bliznashka L, et al.[105] 2021 | Tanzania | Rural; Urban | Mental Health | 593 | CHW | Positive |
| Bolton P, et al.[106] 2014 | Iraq | Rural | Mental Health | 281 | CHW | Positive |
| Buttorff C, et al.[40] 2012 | India | Rural; Urban | Mental Health | 2,796 | CHW + PHC doctors + Psychiatrist | Positive |
| Cajanding RJ, et al.[107] 2016 | Philippines | Unknown | NCD combination | 123 | Nurse | Positive |
| Cajanding RJ, et al.[108] 2017 | Philippines | Urban | CVD | 199 | Nurse | Positive |
| Cakir H, et al.[88] 2006 | Turkey | Urban | CVD | 70 | Nurse | Positive |
| Cal A, et al.[109] 2020 | Turkey | Urban | Cancer | 200 | Nurse | Positive |
| Cappuccio FP, et al.[110] 2006 | Ghana | Rural | CVD | 1013 | CHW | Positive |
| Castle P, et al.[111] 2019 | Brazil | Urban | Cancer | 483 | CHW | Positive |
| Catley D, et al.[112] 2022 | South Africa | Urban | Diabetes | 494 | CHW | Neutral |
| Chang Z, et al.[89] 2020 | China | Rural | CVD | 80 | Nurse | Positive |
| Chao J, et al.[90] 2012 | China | Urban | CVD | 2400 | CHW | Positive |
| Chaowanee,L et al.[113] 2018 | Thailand | Rural | Mental Health | 60 | Nurse | Positive |
| Chatterjee S, et al.[67] 2014 | India | Rural; Urban | Mental Health | 282 | CHW | Positive |
| Chen S, et al.[41] 2015 | China | Urban | Mental Health | 326 | Nurse + doctor + psychiatrist | Positive |
| Chibanda D, et al.[114, 115] 2016 | Zimbabwe | Rural; Urban | Mental Health | 573 | CHW | Positive |
| Dehghan N et al.[116], 2020 | Iran | Urban | Diabetes | 52 | Nurse | Positive |
| DePue J, et al.[117] 2013 | American Samoa | Urban | Diabetes | 268 | Nurse + CHW | Positive |
| de Souza E, et al.[118] 2014 | Brazil | Urban | CVD | 252 | Nurse | Positive |
| de Souza CF, et al.[119] 2017 | Brazil | Urban | Diabetes | 118 | CHW | Neutral |
| Dhoj Shrestha A, et al.[120] 2022 | Nepal | Urban | Cancer | 690 | Female community health volunteer | Positive |
| Dorsey S, et al. [30], 2020 | Tanzania and Kenya | Rural; Urban | Mental Health | 640 | Lay counsellors | Positive |
| Esmaeilpour-BandBoni M, et al. [45], 2021 | Iran | Urban | Diabetes | 66 | Nurse | Positive |
| Fairall L, et al.[91] 2016 | South Africa | Rural | NCD combination | 4,393 | Nurse | Neutral & negative |
| Gamage DG, et al.[121] 2020 | India | Rural | CVD | 1736 | CHW | Positive |
| Gao G, et al.[122] 2020 | China | Urban | Respiratory | 180 | Nurse | Positive |
| Garcia-Pena C, et al. [23], 2001 | Mexico | Urban | CVD | 718 | Nurse | Positive |
| Gaudel P, et al.[123] 2021 | Nepal | Urban | CVD | 224 | Nurse | Positive |
| George C, et al.[124] 2020 | India | Rural | Mental Health | 214 | CHW | Positive |
| Getachew S, et al.[125] 2022 | Ethiopia | Rural | Cancer | 162 | Nurse | Positive |
| Ghanbari E, et al. [46] 2021 | Iran | Urban | Cancer | 82 | Nurse | Positive |
| Ginsburg O, et al. [50] 2014 | Bangladesh | Rural | Cancer | 22,337 | CHW | Positive |
| Goudge J, et al.[76] 2018 | South Africa | Rural | NCD combination | 2508 | Lay Health Workers | Neutral |
| Guerra-Riccio G, et al.[126] 2004 | Brazil | Urban | CVD | 100 | Nurse + Pharmacist | Positive |

*(Continued)*

**Table 1.** (Continued)

| Author, Year | Country | Setting | Disease or condition | Participant number | Workforce included | Intervention impact* |
|---|---|---|---|---|---|---|
| Gul A, et al.[127] 2004 | Pakistan | Urban | Mental Health | 366 | Community counsellors + Doctor + psychiatrist + sociologist + clinical psychologists | Positive |
| Gureje O, et al.[72] 2019 | Nigeria | Rural; Urban | Mental Health | 1178 | CHW + PHC doctor | Neutral |
| Gureje O, et al. [31] 2020 | Nigeria and Ghana | Urban | Mental Health | 307 | PHC providers + Traditional and faith healers | Positive |
| Gyawali B et al.[84] 2021 | Nepal | Urban | Diabetes | 244 | CHW | Positive |
| Hanlon C, et al. [28] 2022 | Ethiopia | Rural | Mental Health | 329 | CHW + PHC doctor | Neutral |
| Hasandokht T, et al.[128] 2015 | Iran | Urban | CVD | 161 | Nurse | Positive |
| He J, et al.[56] 2017 | Argentina | Urban | CVD | 1432 | CHW + PHC doctor | Positive |
| He J, et al.[21] 2023 | China | Rural | CVD | 33995$^\beta$ | Non-physician community health-care providers | Positive |
| Huang Y-J, et al.[129] 2017 | China | Urban | NCD combination | 120 | Nurse | Positive |
| Jafar TH, et al.[74] 2009 | Pakistan | Urban | CVD | 1341 | CHW + PHC doctor | Positive |
| Jafar TH et al.[130] 2010 | Pakistan | Urban | CVD | 4023 | CHW | Positive |
| Jafar TH, et al. [34] 2020 | Bangladesh, Pakistan, and Sri Lanka | Rural | CVD | 2645 | CHW | Positive |
| Jain V, et al.[131] 2018 | India | Rural | Diabetes | 299 | CHW | Neutral |
| Jayasuriya R et al.[132] 2015 | Sri Lanka | Urban | Diabetes | 85 | Nurse | Positive |
| Jeemon P, et al.[133]2021 | India | Urban | CVD | 1671 | Nurse + CHW | Positive |
| Jiamjariyapon T, et al.[42] 2017 | Thailand | Rural | CKD | 442 | Nurse, CHW, Doctor, chronic care nurse, pharmacist, nutritionist, physical therapist, community network teams | Positive |
| Jiang X, et al.[134] 2007 | Turkey | Urban | Respiratory | 61 | Nurse | Positive & neutral |
| Jiang W, et al.[135] 2020 | China | Urban | CVD | 144 | Nurse | Positive & neutral |
| Joshi R, et al.[136–138] 2012 | India | Rural | CVD | 1137 | CHW | Positive |
| Joshi R, et al. [139] 2019 | India | Rural | CVD | 3261 | CHW | Neutral |
| Kaaya SF, et al. [92] 2013 | Tanzania | Urban | Mental Health | 311 | Nurse + Midwife | Positive & neutral |
| Kalani Z, et al.[27] 2022 | Iran | Urban | CVD | 80 | Nurse | Positive |
| Kara M, et al.[140] 2007 | Turkey | Urban | Respiratory | 61 | Nurse | Neutral |
| Kargar JM et al.[85] 2015 | Iran | Urban | Mental Health | 60 | Nurse | Positive |
| Keliat B, et al.[141] 2020 | Indonesia | Urban | Mental Health | 193 | Nurse | Positive |
| Khetan A et al.[142] 2019 | India | Urban | NCD combination | 1242 | CHW | Positive |
| Khezri E, et al. [26] 2022 | Iran | Urban | Cancer | 72 | Nurse | Positive |
| Kondal D, et al.[143] 2022 | India | Rural | CVD | 13813 | CHW (ASHA) | Negative |
| Labhardt ND, et al.[93] 2011 | Cameroon | Rural | NCD combination | 221 | Nurse | Positive |
| Li P, et al.[144], 2015 | China | Urban | Respiratory | 68 | Nurse | Positive |
| Li X, et al.[47], 2020 | China | Urban | Respiratory | 70 | Nurse | Positive |
| Li C, et al.[145] 2023 | China | Urban | Cancer | 178 | Nurse | Positive |
| Liang R, et al.[146] 2012 | China | Rural; Urban | Diabetes | 62 | Nurse | Positive |
| Liu CY, et al.[147] 2010 | China | Rural; Urban | Cancer | 1510 | Nurse | Positive |
| Liu H, et al.[35] 2015 | China | Urban | Diabetes | 128 | Dietitian | Neutral |
| Lund C, et al.[68] 2020 | South Africa | Urban | Mental Health | 425 | CHW | Neutral |
| Ma C, et al.[148] 2014 | China | Urban | CVD | 120 | Nurse | Positive |

*(Continued)*

**Table 1.** (Continued)

| Author, Year | Country | Setting | Disease or condition | Participant number | Workforce included | Intervention impact* |
|---|---|---|---|---|---|---|
| Malakouti SK, et al. [86], 2016 | Iran | Unknown | Mental Health | 241 | CHW | Positive |
| Mash RJ, et al.[149] 2014 | South Africa | Urban | Diabetes | 1570 | CHW | Neutral |
| Mehralian H, et al. [150], 2014 | Iran | Unknown | CVD | 110 | Nurse | Positive |
| Mini GK, et al. [25], 2022 | India | Urban | CVD | 402 | Nurse | Positive |
| Mittra I, et al.[151] 2021 | India | Urban | Cancer | 151538 [π] | CHW | Positive |
| Mollaoğlu M, et al.[18] 2009 | Turkey | Urban | Diabetes | 50 | Nurse | Positive |
| Moreira R, et al.[152] 2015 | Brazil | Rural | Diabetes | 80 | Nurse | Positive |
| Muchiri JW, et al.[36] 2016 | South Africa | Rural | Diabetes | 82 | Dietitian | Neutral |
| Myers B, et al.[153] 2022 | South Africa | Urban; Rural | Mental Health | 1340 | CHW | Positive |
| Neupane D, et al.[154] 2018 | Nepal | Urban | CVD | 1638 | CHW | Positive |
| Ogedegbe G, et al.[155, 156] 2018 | Ghana | Urban | CVD | 757 | Nurse | Positive |
| Okube OT, et al.[157] 2023 | Kenya | Urban | CVD | 352 | Nurse | Positive |
| Osborn TL, et al. [158], 2021 | Kenya | Unknown | Mental Health | 413 | High school graduates | Positive |
| Pace L, et al.[159] 2018 | Rwanda | Rural | Cancer | 1801 | Nurse + CHW | Positive |
| Pan Y et al.[160] 2019 | China | Unknown | Mental Health | 112 | Nurse | Positive |
| Patel V, et al.[43, 161] 2010 | India | Rural; Urban | Mental Health | 2796 | CHW + PHC doctors + Psychiatrist | Positive |
| Peiris D, et al. [54] 2019 | India | Rural | NCD combination | 8,642 | CHW + PHC doctor | Positive |
| Petersen I, et al.[162] 2021 | South Africa | Urban | NCD combination | 1043 | Nurse + Lay counsellors | Positive |
| Pisani P, et al.[163] 2006 | Philippines | Urban | Cancer | 1293 | Nurse | Neutral |
| Prabhakaran D, et al. [53] 2018 | India | Rural | NCD combination | 3698 | Nurse + Doctor | Neutral |
| Pradeep J, et al.[164] 2014 | India | Rural | Mental Health | 260 | CHW | Positive & neutral |
| Rahman A, et al. [165] 2008 | Pakistan | Rural | Mental Health | 903 | Lady Health Workers | Positive |
| Rahman A, et al. [166] 2016 | Pakistan | Unknown | Mental Health | 346 | CHW | Positive |
| Rahul A, et al.[167] 2021 | India | Rural; Urban | Diabetes | 132 | Nurse | Positive |
| Rylance S, et al.[168] 2021 | Malawi | Urban | Respiratory | 120 | CHW | Positive |
| Saffi M, et al.[169] 2014 | Brazil | Urban | CVD | 80 | Nurse | Positive |
| Safren SA, et al. [170] 2021 | South Africa | Unknown | Mental Health | 163 | Nurse | Positive |
| Saisanan Na Ayudhaya W, et al.[171] 2020 | Thailand | Rural | Mental Health | 82 | Nurse + CHW | Positive |
| Salimzadeh H, et al.[172,173] 2018 | Iran | Unknown | Cancer | 122 | Nurse | Positive |
| Samonnan T, et al.[174] 2018 | Thailand | Unknown | NCD combination | 93 | Nurse + Researcher | Positive |
| Sankaranarayanan R, et al.[175] 2007 | India | Rural | Cancer | 80269 | Nurse + CHW + Doctor | Positive |
| Sartorelli D, et al.[37] 2005 | Brazil | Urban | Diabetes | 104 | Nutritionist | Positive |
| Scain SF, et al.[176] 2009 | Brazil | Rural | Diabetes | 104 | Nurse | Positive |
| Scazufca M, et al.[177] 2022 | Brazil | Urban | Mental Health | 715 | CHW | Positive |
| Schwalm JD, et al.[29] 2019 | Colombia; Malaysia | Rural; Urban | CVD | 1371 | CHW | Positive |
| Secginli S, et al.[178] 2011 | Turkey | Urban | Cancer | 216 | Nurse | Positive & neutral |
| Selvaraj F, et al.[179] 2012 | Malaysia | Rural | CVD | 297 | Nurse | Positive |
| Sharma KK, et al. [180], 2016 | India | Unknown | CVD | 100 | CHW | Positive |
| Shastri SS, et al.[20] 2014 | India | Urban | Cancer | 151538 [π] | Primary Health care workers | Positive |
| Shelley D, et al.[181] 2021 | Vietnam | Unknown | CVD | 1312 | CHW | Positive |
| Shi Y, et al.[182] 2020 | China | Unknown | NCD combination | 100 | Nurse | Positive |
| Siabani S, et al.[183] 2016 | Iran | Urban | CVD | 231 | Doctor + community health volunteer | Positive |
| Sinha B, et al.[184] 2021 | India | Rural; Urban | Mental Health | 1950 | Trained study workers | Positive |

*(Continued)*

**Table 1.** (Continued)

| Author, Year | Country | Setting | Disease or condition | Participant number | Workforce included | Intervention impact* |
|---|---|---|---|---|---|---|
| Steffen PLS et al.[185] 2021 | Brazil | Unknown | NCD combination | 189 | Nurse | Positive |
| Subramanian SC, et al.[186] 2020 | India | Urban | Diabetes | 70 | Nurse | Neutral |
| Sun J, et al.[187] 2008 | China | Urban | Diabetes | 150 | Nutritionist | Positive |
| Sun Y, et al.[22] 2022 | China | Rural | CVD | 33995 [β] | Village doctor | Positive |
| Temucin E, et al.[188] 2018 | Turkey | Unknown | Cancer | 110 | Nurse | Positive |
| Thakur D, et al.[189] 2019 | India | Urban | NCD combination | 80 | Nurse | Positive |
| Tian M, et al.[33] 2015 | China, India | Rural | CVD | 2,086 | CHW | Positive |
| Tomlinson M, et al.[190] 2018 | South Africa | Urban | Mental Health | 1238 | CHW | Neutral |
| Vedanthan R, et al.[51] 2019 | Kenya | Rural | CVD | 1460 | CHW | Positive |
| Wagner G et al.[191] 2016 | Uganda | Urban | NCD combination | 1252 | Nurse | Positive |
| Wang Y, et al.[192] 2014 | China | Urban | Respiratory | 92 | Nurse | Positive |
| Wang LH, et al.[193] 2020 | China | Unknown | Respiratory | 154 | Nurse | Positive |
| Wang G, et al.[194] 2021 | China | Urban | NCD combination | 168 | Nurse | Positive |
| Weiss WM, et al.[195] 2015 | Iraq | Unknown | Mental Health | 149 | CHW | Positive |
| Wroe EB, et al.[196] 2021 | Malawi | Rural | NCD combination | >29000[a] | CHW | Positive & Neutral |
| Xavier D, et al.[197] 2016 | India | Urban | CVD | 806 | CHW | Positive |
| Xu DR, et al.[52] 2019 | China | Rural | Mental Health | 278 | Lay Health Workers | Positive |
| Xueyu L, et al.[198] 2015 | China | Urban | CVD | 77 | Advanced practice nurse | Positive & neutral |
| Yan H, et al.[44] 2021 | China | Urban | CVD | 249 | Nurse + cardiac nurse, cardiologist, electrophysiologist, psychologist, physiotherapist | Positive & neutral |
| Yin Z, et al.[199] 2018 | China | Rural | Diabetes | 184 | CHW | Neutral |
| You J, et al.[200] 2020 | China | | CVD | 152 | Nurse | Neutral |
| Yuan X, et al.[201] 2015 | China | Rural | Respiratory | 1008 | Nurse | Positive |
| Pan Y, et al.[202] 2019 | China | | Mental Health | 112 | Nurse | Positive |
| Zhang P, et al.[203] 2017 | China | Urban | CVD | 236 | Nurse | Positive |
| Zheng X, et al.[204] 2020 | China | | CVD | 173 | Nurse | Positive |
| Zhu X, et al.[48] 2018 | China | Urban | CVD | 134 | Nurse | Positive |

CHW: community health workers; CVD: cardiovascular disease; PHC: Primary Health Care; [a]Cluster Randomised Controlled Trial at household level, where 29000 households were involved.

*Intervention impact relates to the primary outcome result when comparing intervention to the control; [β]Articles used same sample; [π]Articles used same sample.

## Uses of digital health in task-sharing

The intervention involved digital health solutions in 15 studies. Four of the digital health studies included nurses[45–48] and six included CHWs[29, 33, 49-51] or lay health workers[52]. Three studies had a team of a doctor with either CHW or nurse, and one study involved a multidisciplinary team of five specialists and a nurse[44]. Twelve out of the 15 studies included management of the condition[29,33,44,48–56], one focussed on rehabilitation[57], one on screening[58], and one focussed on education[46].

Digital health interventions included the use of clinical decision support tools and electronic health records to help the NPHW with diagnosis, treatment and referral[29,33,49,51,53,54]. In one study, a smartphone application was used by the patients to set reminders to improve medicine adherence[24]. Smartphone applications were also used to train CHWs[49, 50, 54], to enable online patient support groups[46] and to correspond with patients[24,52].

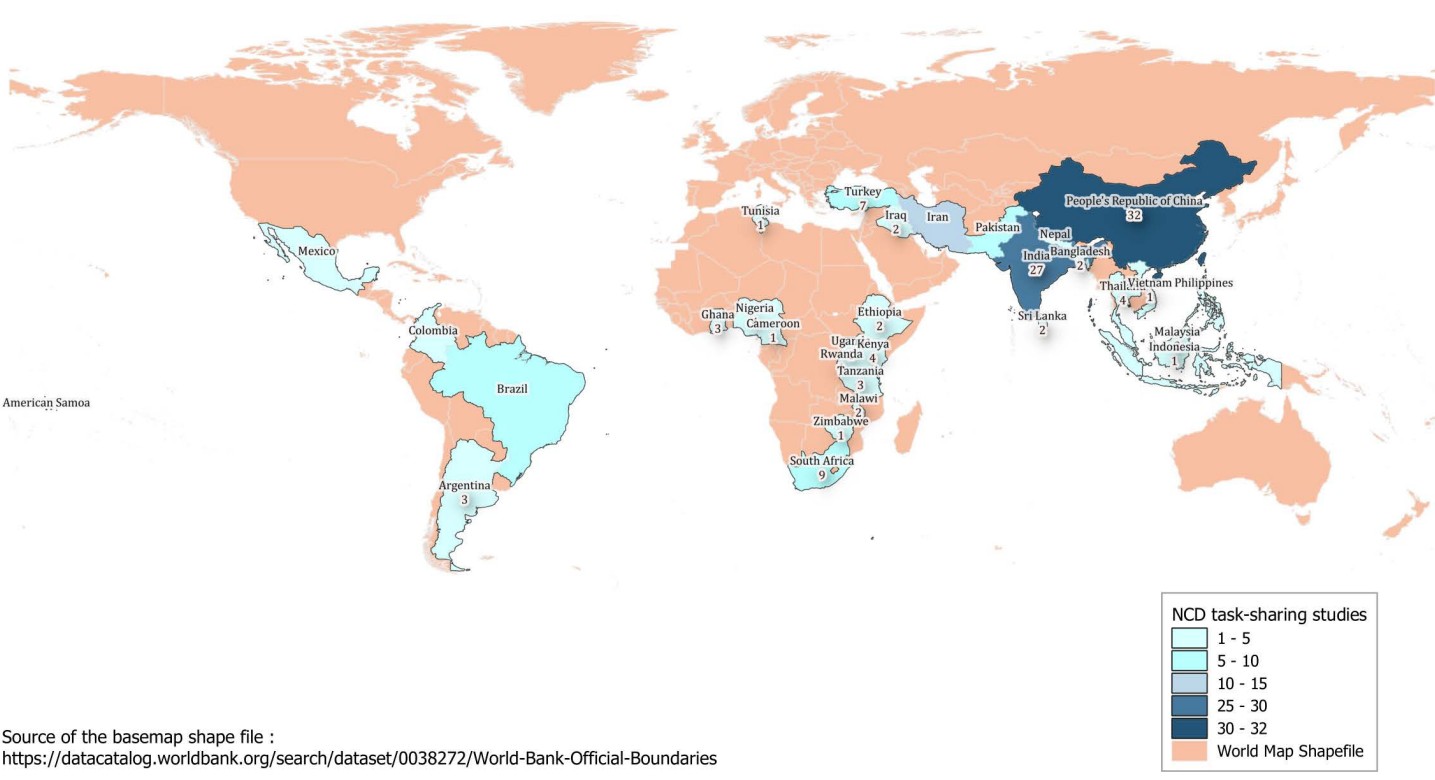

**Fig 2. Number of studies extracted in each country. Source of the basemap shape file:** https://datacatalog.worldbank.org/search/dataset/0038272/World-Bank-Official-Boundaries.

## Diseases addressed by task-sharing interventions

The diseases or conditions investigated varied between the studies (Table 1). Most studies focussed on cardiovascular disease or its risk factors (32%, 47/149). Of these, studies specifically focused on hypertension (43%, 30/47), ischaemic heart disease (15%, 7/47), heart failure (13%, 6/47), and CVD risk factors in general (32%, 15/47). The second most common condition studied was mental health (32%, 48/149) which included depression, anxiety, and post-traumatic stress. eighteen percent of all studies focussed on diabetes (27/149). Cancer was investigated in 13% (20/149) of the included studies. Of these, interventions specifically focused on breast cancer (55%, 11/20), cervical cancer (30%, 6/20), and gastrointestinal cancer (15%, 3/20). Respiratory conditions included asthma and chronic obstructive pulmonary diseases (7%,10/149). One study focussed on the management of chronic kidney disease (1/149). Seventeen studies (11%) included two or more conditions, and 13 of this included mental health along with another chronic illness.

## Was task-sharing effective in improving health outcomes?

Out of the 149 studies, 120 studies (81%) reported a significant primary outcome (or at least 1 significant primary outcome), while 19 studies reported neutral results, and one reported a negative result. Table 1 highlights the effect of each outcome. All the task-sharing studies involving the care for cancer morbidity reported at least one positive primary outcome. Twelve of the 15 task-sharing studies which used a digital health intervention reported at least one positive primary outcome, while three reported non-statistically significant outcomes[48,54,59]. Outcomes of all task-sharing interventions are shown on Table 2. In

**Table 2. Characteristics and outcomes of the interventions.**

| Author, Year | Disease | Activity | Management details for MLHP | Workforce | Primary outcome | Primary outcome result |
|---|---|---|---|---|---|---|
| Adewuya AO, et al.[38] 2019 | Mental health: depression | Management, Promotion, Rehabilitation | CHW: provide enhanced Psychoeducation, including symptom education, an association between depression with interpersonal difficulties.<br>Nurse: deliver problem-solving therapy<br>Doctor: initiate and maintain anti-depressants medication<br>Clinical support and supervision from the mental health team* | Doctor, nurse, CHW | Recovery score | Recovery rate in the intervention vs control group - 60.3% vs 18.2% (ARR 3.10, 95% CI 2.15 - 3.87) |
| Adeyemo A, et al.[87] 2013 | CVD: Hypertension | Management, Promotion | Nurse: facilitation of clinic visits and health education.<br>Treatment was implemented by a nurse with physician backup (for dose titration). | Nurse, Doctor | BP lowering medication adherence | Overall, ~77% of participants took > 98% of prescribed pills. Adherence did not differ by treatment arm |
| Al Ksir K, et al.[100] 2022 | Diabetes | Management | Nurse-structured motivational interviewing to support youth with type-1 diabetes on their personal development and on self-management skills for their condition. Also empowered them with skills for behaviour change. | Nurse | Changes in TRAQ (Transition Readiness Assessment Questionnaire) scores | Significant differences in the mean TRAQ scores (3-months: 3.53 (sd=0.56) vs 2.11 (sd=0.57); 6-months: 4.25 (sd=0.38) vs 2.31 (sd=0.50); p<0.001) in the intervention group compared to the control group. |
| Ali M, et al.[39] 2020 | Diabetes; Mental Health | Management, Promotion | nonphysician care coordinators (who had background on nutritional counseling, and social work): provide patient-level self-management, such as adherence to diet plans, exercise, and medications, of diabetes and depression.<br>Diabetologist: initiate and/or modify behavioural or pharmacotherapies interventions for depression, glucose, blood pressure, and lipid management. | Nonphysician care coordinator, diabetologist | Depression symptoms and cardiometabolic indices | A significantly greater percentage of patients in the intervention group vs the usual care group met the primary outcome (71.6% vs 57.4%; risk difference, 16.9% [95% CI, 8.5%–25.2%] |
| Arjunan P, et al.[101] 2021 | CVD: Heart Failure | Management, Promotion, Rehabilitation | Monitoring patient progress and outcomes | Nurse | Quality of life | Improvement in the physical component (t=2.23,p=.02), mental component (t= 11.17,p<.001),and disease-specific (t= 5.92,p<.001) |
| Arrossi S, et al.[102] 2015 | Cancer: Cervical cancer | Prevention, Promotion | Distribution of cervical sample kits to CHWs. Collecting and managing data on population screening.<br>Colposcopy and biopsy for screen-positive patients. | CHW | Screening uptake | Increase in the proportion of women having HPV test per CHW allocated to the intervention (86%) versus (20%) in the control group (p<0·0001). |
| Azami G, et al.[103] 2018 | Diabetes | Management, Promotion | Behaviour change (motivational interview), healthy eating, being active, monitoring of blood glucose, taking medication, foot care, reducing risk, and healthy coping. Receiving telephone follow-up calls. | Nurse | HbA1c | Intervention group had significantly lower HbA1c values (47.9%) than those in the control group. At week 24, the differences increased to 62% (P<0001). |
| Bass J, et al.[104], 2016 | Mental Health | Promotion, Management | Psychosocial support and psychoeducation, compassionate care, treatment planning, and medication management. | CHW | Depressive and dysfunction symptoms | Intervention had a significant and moderate-sized effect on depression symptoms (P= 0.02) and dysfunction (P= 0.03) |

*(Continued)*

**Table 2.** (Continued)

| Author, Year | Disease | Activity | Management details for MLHP | Workforce | Primary outcome | Primary outcome result |
|---|---|---|---|---|---|---|
| Beratarrechea A, et al. [49], 2019 | CVD: CVD risk | Promotion, Screening, Prevention | mhealth tools used for screening and appointment scheduling | CHW | Attendance | 49.4% of the participants with moderate or high risk of CVD completed a first visit to PHCs within 6 weeks of being screened by a CHW, compared with 13.5% in the control group. |
| Bliznashka L, et al. [105] 2021 | Mental health | Promotion, Prevention, Promotion | Non-pharmacological therapy (e.g. developmentally appropriate stimulation of children, education on early childhood development, etc.), counselling, and promotion of caregiver responsiveness, nutrition education. | CHW | Depression and anxiety symptoms | The pooled intervention arms significantly reduced depressive symptoms and anxiety sub-scores as compared with control. |
| Bolton P, et al. [106] 2014 | Mental health | Prevention; Management; Rehabilitation | Cognitive processing therapy (identifying, challenging and modifying maladaptive beliefs), Behavioural activation treatment for depression. | CHW | Reduction of depression symptoms | Depression effect sizes were 0.60 and 0.84 for BATD and 0.70 and 0.44 for CPT compared to all controls and BATD/CPT-specific controls respectively. Effect sizes were statistically significant |
| Buttorff C, et al. [40] 2012 | Mental health | Screening; Management; Referral | Psychosocial care, treatment prescription, review of treatment and dosage adaptation, interpersonal therapy (relationships with others and coping strategies). | CHW | Recovery from common medical disorders | Modest evidence of an effect of the intervention on recovery (proportion recovered: 65.0% vs. 52.9% in intervention and enhanced usual care arms respectively; RR=1.22, 95%CI 1.00,1.47) |
| Cajanding RJ, et al. [107] 2016 | Heart Failure; Mental health | Prevention; Management | Cognitive behavioural therapy (patient education, self-monitoring, skills training, cognitive restructuring and spiritual development), treatment administration. | Nurse | Quality of life, self-esteem and mood among patients living with Heart failure | After the 12-week intervention period, participants in the intervention group had significant improvement in their quality of life, self-esteem and mood scores compared with those who received only standard care. |
| Cajanding RJ, et al. [108] 2017 | CVD: Ischaemic heart disease | Prevention; Promotion, Management | Overseeing structured discharge planning program | Nurse | Patients' perceived functional status | Significant difference in perceived functional status scores between the control and the intervention groups of 8.59 +/- 2.29 (95%CI, 4.02–13.16; p<0.01). |
| Cakir H, et al. [88] 2006 | CVD: Hypertension | Management | Education classes and counselling sessions | Nurse | Blood Pressure | Both SBP and DBP decreased in the intervention group but not in the control group. Mean reductions in SBP and DBP were 8.8 (SD= 5.2) and 6.9 (SD= 5.3) mmHg, respectively. |
| Cal A, et al. [109] 2020 | Cancer: Breast cancer | Prevention; Promotion, Management | Deliver lymphedema prevention lessons | Nurse | Incidence of lymphedema | 17.1% of the control group had signs of lymphedema, and none in the intervention group had lymphedema (p<.05) |
| Cappuccio FP, et al. [110] 2006 | CVD: Hypertension | Prevention; Promotion | Health education | CHW | Urinary sodium excretion | Non-significant change in sodium excretion |

*(Continued)*

**Table 2.** (Continued)

| Author, Year | Disease | Activity | Management details for MLHP | Workforce | Primary outcome | Primary outcome result |
|---|---|---|---|---|---|---|
| Castle P, et al.[111] 2019 | Cancer: Cervical Cancer | Promotion, Screening | Identifying women eligible for screening & assigning these to CHWs. Colposcopy and biopsy for screen-positive patients. Processing samples collected. | CHW | Screening uptake | Women were significantly more likely to adhere to screening when undertaking self-collection & HPV testing compared to Pap testing (p<0.001). |
| Catley D, et al.[112] 2022 | Diabetes and CVD | Prevention | Behaviour change, strategies for setting goals on lifestyle: self-monitoring of physical activity and diet, healthy nutrition, stress management, and stimulus control. | CHW | Percentage of weight loss | Percentage of weight changes from baseline (intervention=-0.61 (95% confidence interval: -1.22 to 0.01), control: -0.44 (95% confidence interval: -1.06 to 0.18); p=) and the group differences were not statistically different (-0.17 (95% confidence interval: -1.04 to 0.71); p=71). |
| Chang Z, et al.[89] 2020 | Multiple: Ischemic heart disease; Mental health | Management, Promotion | Percutaneous coronary intervention, counselling, cognitive behavioural therapy | Nurse | Cardiac outcomes, quality of life and mental health status | Significant increase in the scores of the 3 domains of Angina Questionnaire in the intervention group (P <.01). Mental health and physical health scores also increased (P <.01). |
| Chao J, et al.[90] 2012 | CVD: Hypertension, Mental health | Prevention | Diet advice, psychological aspects of health, tailor-made exercise program, training on health self-management | CHW | Knowledge, psychological condition, BP monitoring, Waist-hip ratio, SBP, DBP | Improvement in (P<0.01): health knowledge score, psychological conditions, physical activity duration, BP monitoring, waist-to-hip ratio, SBP and fasting blood sugar |
| Chaowanee,L et al.[113] 2018 | Mental health | Management, Promotion | Psychoeducation, advice to exercise. | Nurse | Mean depression scores | Mean depression scores of the intervention group were significantly low at end of study. |
| Chatterjee S, et al.[67] 2014 | Mental health | Prevention; Promotion; Management | Education, treatment, prescription, consultations, information about illness | CHW | Reduction in symptoms and disability | Participants in the intervention group had lower scores across all subdomains of symptoms and disabilities compared to those in the control group. |
| Chen S, et al.[41] 2015 | Mental health | Promotion; Screening; Management; Referral | Primary care nurse: consultation, treatment prescription, screening, education about illness and treatment adherence, referral Doctor: conduct depression diagnosis during screening stage, depression treatment and informed of each patient's health questionnaire Psychiatrist: supervise and support collaborative team function | Primary care nurse, Doctor, Psychiatrist | Reduction in depression score | Patients to intervention had a significantly greater reduction in scores than did those in practices assigned to enhanced care as usual. |
| Chibanda D, et al. [114, 115] 2016 | Mental health | Prevention; Management | Problem-solving therapy (a structured approach to identifying problems and finding workable solutions), screening, referral. | CHW | Common mental disorders | The primary outcome of SSQ-14 scores for common mental disorders was lower in the intervention group than in the control group (mean, 3.81; 95% CI, 3.28 to 4.34 vs 8.90; 95% CI,8.33 to 9.47 |

*(Continued)*

**Table 2.** (Continued)

| Author, Year | Disease | Activity | Management details for MLHP | Workforce | Primary outcome | Primary outcome result |
|---|---|---|---|---|---|---|
| Dehghan N et al.[116], 2020 | Diabetes | Promotion, Management, Referral | Nurse: Monitoring and evaluation of wounds, wound dressing, and blood glucose control. Coordination and management of specific cases, provision of basic care, patient education and self-care, patient referral, wound debridement, and treatment of bone abnormalities. | Nurse led team | Quality of care and HbA1C | Significant difference in quality-of-life score and HbA1c between intervention and control groups (P <.0001). |
| DePue J, et al.[117] 2013 | Diabetes | Management | Nurse: Group education sessions, communicate with physicians about patient care needs, CHWs: appointments adherence, diabetes education, adherence to medication, problem-solving, support for self-management, referral to primary care | Nurse, CHW | HbA1c levels | At 12 months, mean HbA1c was significantly lower among participants, compared with usual care, after adjusting for confounders (b=20.53; SE = 0.21; P= 0.03). |
| De Souza E, et al.[118] 2014 | CVD: Heart Failure | Management, Promotion | Monitoring patient progress and outcomes | Nurse | First visit to the emergency, readmission, or all-cause death | The composite endpoint of a first HF-related visit to the emergency department, hospital readmission, or all-cause death was decreased in the interventional group [relative risk 0.73; P=0.049]. |
| De Souza CF, et al.[119] 2017 | Diabetes | Management, Promotion | Diabetes education (understanding the disease, medication adherence) | CHW | HbA1c | Significant reduction in HbA1c levels in both the groups (intervention: 9.1±2.2 vs. 7.9±1.9%; control: 9.1Â2.1 vs. 8.4±2.5%, p overtime<0.001) |
| Dhoj Shrestha A, et al.[120] 2022 | Cancer: Cervical Cancer | Screening | Health education, counselling, repeat visits, referral for visual inspection of the cervix with acetic acid. | FCHV | Change in cervical cancer screening | There was a significant increase in cervical cancer screening uptake in the intervention group compared to the control group (RR=1.48; 95% confidence interval: 1.32 to 1.66; p<0.0001) |
| Dorsey S, et al. [30], 2020 | Mental Health | Rehabilitation | Cognitive Behavioural Therapy focusing on grief specific elements, Psychoeducation | Lay counsellors | Post traumatic stress symptoms | Compared with usual care, the intervention was more effective both after treatment and at 12 months in improving PTS in children. |
| Esmaeilpour-BandBoni M, et al. [45], 2021 | Diabetes | Management, Promotion | Treatment adherence, Education about disease, Lifestyle education (diet, self-care and exercise) | Nurse | HbA1c | Decrease in HbA1c in the intervention group was significantly greater than that of the control group (P<0.001). |
| Fairall L, et al.[91] 2016 | CVD: Hypertension; Diabetes; Respiratory; Mental health | Management | Educational outreach, treatment prescription, counselling, screening, review treatment/dose adjustment. | Nurse | Treatment intensification | Treatment intensification rates in intervention clinics were not superior to those in the control clinics |
| Gamage DG, et al.[121] 2020 | CVD: Hypertension | Management, Promotion | Measurement of blood pressure, weight, waist circumference, education about diet, physical activity and hypertension | CHW | Blood pressure | 1.6-fold (95%CI: 1.2-2.1; p-value=0.001) better control of hypertension in the intervention group than in the control group. |

*(Continued)*

**Table 2.** (Continued)

| Author, Year | Disease | Activity | Management details for MLHP | Workforce | Primary outcome | Primary outcome result |
|---|---|---|---|---|---|---|
| Gao G, et al.[122] 2020 | Respiratory: Asthma | Promotion; Management, Prevention | Nurse: provide standard care with structured education and skills training for patients in a continuous process (outpatient sessions, home visit and telephone visit) | Nurse | Asthma control and quality of life | Significant difference (P=.002) in the asthma control score between intervention and control group. Significant difference in health-related quality of life scores between the groups. |
| Garcia-Pena C, et al.[23], 2001 | CVD: Hypertension | Prevention; Promotion, Management | Measured blood pressure, reviewed baseline information, discussed lifestyle changes. Reviewed pharmacological treatment, provided adherence support. | Nurse | Blood pressure | 36.5% of the intervention versus 6.8% of the control group had a BP of <160/90 mmHg. |
| Gaudel P, et al.[123] 2021 | CVD: Ischaemic heart disease | Prevention; Management; Promotion | Screening eligible patients, monitoring outcomes | Nurse | Lifestyle-related risk factors | Five out of the seven studied lifestyle-related risk factors differed significantly between the study groups: diet, adherence to medication, perceived stress, and smoking and alcohol consumption. |
| George C, et al.[124] 2020 | Mental health | Prevention; Promotion | Cognitive Behavioural Therapy, screening | CHW | Postnatal depression | 30% reduced prevalence of depression in the active intervention group compared to the control groups. Not statistically significant. |
| Getachew S, et al.[125] 2022 | Cancer: Breast cancer | Management | Health education and literacy material, empathetic counselling, phone call reminders, monitoring of medication side effects and compliance. | Nurse | Adherence to Tamoxifen | Adherence was found to be 90% (36/40) in the intervention group and 79.3% (23/29) in the control group (p=0.302). |
| Ghanbari E, et al. [46] 2021 | Cancer: Breast cancer; Mental health | Promotion; Management | Training nurses to deliver support sessions | Nurse | Anxiety and self-esteem | Statistically significant differences between the groups 1 week after completing the intervention (P<.001). |
| Ginsburg O, et al. [50] 2014 | Cancer: Breast cancer | Promotion; Screening; Referral | Deliver educational intervention. | CHW | Clinic attendance | Adherence was high in all three study arms. Adherence was highest for women interviewed by CHWs with smartphones and who acted as patient navigators. |
| Goudge J, et al.[76] 2018 | CVD: Hypertension | Management, Promotion | LHWs: assist with booking appointments, retrieving and filing patient files, and providing health education (on adherence and lifestyle). Measure vital signs, assist the nurses with the prepacking of medications. Nurses: diagnoses, prescribes, and dispenses medication | Lay health workers, nurses | Blood pressure | No improvement in BP control among users of intervention clinics as compared with control clinics. |
| Guerra-Riccio G, et al.[126] 2004 | CVD: Hypertension | Management, Promotion | Pharmacist: visited patients to deliver antihypertensive drugs and perform a pill count. Nurses: adherence support, blood pressure monitoring | Nurse, pharmacist | Blood pressure | BP declined more in intervention than in control arm (35 ± 5/19 ± 3 v 27 ± 5/9 ± 3 mm Hg). |
| Gul A, et al.[127] 2004 | Mental health | Management | Psychotherapy/Cognitive Behavioural Therapy, Counselling. | Community based counsellors | Anxiety and/or depression | Net reduction of anxiety score in the intervention group of 21% (P=0.001) immediately after 8 weeks of counselling. |

(Continued)

**Table 2.** (Continued)

| Author, Year | Disease | Activity | Management details for MLHP | Workforce | Primary outcome | Primary outcome result |
|---|---|---|---|---|---|---|
| Gureje O, et al.[72] 2019 | Mental health | Promotion; Management; Rehabilitation | Front-line primary care providers: psychoeducation and counselling, screening Doctor: supervision and support clinical supervisions, support, clinical emergency calls, administrative management | Front-line primary care providers (nurses, community health officers, and community health extension workers), Doctors | Remission of depression | No difference in terms of primary outcome (remission at 12 months: 76% in the intervention group vs 77% in the control group) |
| Gureje O, et al.[31] 2020 | Mental health | Management | CHW: provide clinical support to patients, information on best clinical practices to traditional faith healers, and psychoeducation to both patients and caregivers Traditional faith healers: provide routine treatments, including herbal, ritual, and psychosocial interventions | CHW, traditional faith healers | Psychotic symptom | Participants in the intervention arm achieved a significantly better primary outcomes at 6 months than controls |
| Gyawali B et al.[84] 2021 | Diabetes | Management, referral, Promotion | Health promotion counselling, blood glucose monitoring, referral, follow-up for medication adherence | CHW | Mean fasting blood glucose | Significantly greater reduction in intervention than control group (p<0.001) |
| Hanlon C, et al.[28] 2022 | Mental health | Management, referral, Promotion | Nurse/psychiatric nurse/: out-patient psychiatric care CHW: psychosocial education, support people with severe disorder, follow up for people who dropped out of care Community-based lay project worker: augment engagement in care | Nurse, CHW, community-based lay project workers | Clinical psychiatric symptom severity | No evidence of inferiority of task-shared care compared with intervention. The mean score was 27.7 for task-shared care and 27.8 for control |
| Hasandokht T, et al.[128] 2015 | CVD: Hypertension and mental health | Management | Lifestyle education (healthy dietary habits, exercise), psychosocial education | Nurse | BP (mmHg) changes | Participants had significant changes in systolic and DBP, weight, waist circumference, body mass index (BMI), energy, NaCl, and perceived stress scale |
| He J, et al. [56] 2017 | CVD: Hypertension | Promotion; Management | CHW: health coaching, home BP monitoring, and BP audit and feedback Doctor: hypertension treatment | CHW, Doctor | Blood pressure | SBP reduction of 19.3 mm Hg (95% CI, 17.9–20.8 mm Hg) for the intervention group and 12.7 mm Hg (95% CI, 11.3–14.2 mm Hg) for the control group |
| He J, et al.[21] 2023 | CVD: Hypertension | Management | Treatment initiation, ensure appropriate dosage for each patient, provision of discounted or free anti-hypertensive medication, health coaching on home blood pressure monitoring, medication adherence and lifestyle changes. | CHW | Myocardial infarction, stroke, heart failure requiring hospitalisation or resulting in death in a 36-month period | During a median of 36.8 months, the primary outcome was confirmed in 808 participants (1·62% rate per year) in the intervention group and 1127 participants (2.40% rate per year) in the usual care group (hazard ratio with intervention 0·67; 95% confidence interval: 0·61–0·73; p<0·0001) and this was statistically significant. |

(Continued)

**Table 2.** (Continued)

| Author, Year | Disease | Activity | Management details for MLHP | Workforce | Primary outcome | Primary outcome result |
|---|---|---|---|---|---|---|
| Huang Y-J, et al.[129] 2017 | CVD: Ischaemic heart disease | Prevention, Management, Promotion | Education and coaching of patients | Nurse | CHD risk factors - or do you want it to specify which ones | Compared with the control group, the intervention group had a 5 mmHg greater reduction in SBP (t = 2.01, p =.047), larger declines in glucose (t = −2.44, p =.015), cholesterol (t = −2.49, p =.015), body mass index (t = −2.58, p =.011), and depression (t = −2.05, p =.043). No significant group differences in smoking behaviour. |
| Jafar TH, et al.[74] 2009 | CVD: Hypertension | Management; Promotion | CHWs: Family-based home health education Doctors: hypertension management | CHW, Doctor | Blood pressure | Decrease in SBP was significantly greater in the home health education and GP group (10.8 mm Hg [95% CI, 8.9 to 12.8 mm Hg]) than in the GP-only, home health education-only, or no intervention groups (5.8 mm Hg [CI, 3.9 to 7.7 mm Hg] in each; P < 0.001). |
| Jafar TH et al.[130] 2010 | CVD: Hypertension | Management; Promotion | Health education | CHW | Blood pressure | SBP increased in the control group by 1.5mmHg and did not change appreciably (0.1mmHg) in the intervention. Difference was statistically significant (P=0.02) |
| Jafar TH, et al, [32, 34] 2020 | CVD: Hypertension | Management; Promotion | CHWs: home visits for blood-pressure monitoring and counselling GPs: hypertension management | CHW Doctors | Blood pressure | Mean reduction in SBP was 5.2 mm Hg greater in the intervention group than in the control group (95% CI, 3.2 to 7.1; P<0.001). |
| Jain V, et al. [131] 2018 | Diabetes | Prevention; Management | Routine fasting and post-prandial capillary glucose monitoring, Anthropometric measurements, Reinforced evidence-based prescription, lifestyle education (smoking cessation, importance of drug refilling and adherence, physical activity, dietary changes) | CHW | Fasting blood sugar, post-prandial blood sugar, glycosylated haemoglobin, lipid profile, blood pressure | No statistical difference between the intervention and the control group. |
| Jayasuriya R et al. [132] 2015 | Diabetes | Management; Promotion | Medications log, a contract for change document, lifestyle changes (reduce total energy intake, by reducing quantity of starchy foods replaced by vegetables; physical activity promoting culturally appropriate exercise during household work and walking) | Nurse | HbA1c | Significantly lower HbA1c in intervention than control group (p=0.035) |
| Jeemon P, et al. [133] 2021 | CVD risk factors, hypertension, diabetes | Prevention; Promotion; Screening | Health education, BP measurement, glucose measurement using POC devices, follow-up visits | CHW | Achievement or maintenance of any three: BP <140/90 mm Hg, fasting plasma glucose <110 mg/dL, fasting plasma glucose <110 mg/dL, low-density lipoprotein cholesterol <100 mg/dL, abstinence from tobacco | 64% participants in the intervention group and 46% participants in the usual care group achieved the primary outcome. The odds of achieving the primary outcome were two times higher in the intervention group than in the usual care group (OR 2·2, 95% CI 1·7–2·7; p<0·00010) |

*(Continued)*

Table 2. (Continued)

| Author, Year | Disease | Activity | Management details for MLHP | Workforce | Primary outcome | Primary outcome result |
|---|---|---|---|---|---|---|
| Jiamjariyapon T, et al. [42] 2017 | Other: Chronic Kidney Disease | Promotion; Management | Nurse: group counselling during hospital visit CHW: home visits to monitor compliance with the treatment. | Nurse, CHW | Mean eGFR | The mean difference of eGFR overtime in the intervention group was significantly lower than the control group by 2.74ml/min/1.73 m2(95%CI 0.60-4.50,p= 0.009). |
| Jiang X, et al. [134] 2007 | CVD: Ischaemic heart disease | Promotion; Management; Rehabilitation | Supervision and coaching of patients for cardiac rehabilitation | Nurse | Change of health behaviours and physiological parameters | Patients in the intervention group demonstrated a significantly better performance in walking, diet and medication adherence; a significantly greater reduction in serum lipids and significantly SBP, DBP control. The effect of the programme on smoking cessation, body weight, serum high-density lipoprotein, was not confirmed. |
| Jiang W, et al.[135] 2020 | CVD: Ischaemic heart disease | Management; Promotion | Physician: disease assessment, drug adjustment, health education and consultation. Nurse and other team members: adaptation of the healthcare plan and outcome measurement | Nurse, doctor, dietitian, physiotherapist | Self-management behaviours | Significant difference between the intervention and control groups on the disease medical management |
| Joshi R, et al. [136–138] 2012 | CVD: CVD risk | Prevention; Promotion; Screening; Management | CHWs: Screening of patients with CVD, education about diet, tobacco use and physical activity; recommended treatment, referral to doctor and follow up Doctor: prescribing medicines | CHW, Doctor | Proportion of high-risk individual identified | Proportion of high-risk individuals reporting that they were screened for CVD was 12% higher (intervention villages, 63.4% vs. control villages, 51.4%; p0.026) |
| Joshi R, et al. [139]2019 | CVD: CVD risk | Prevention; Promotion | Advising dose escalation for suitable patients | CHW | Blood pressure | Significant decline in SBP (mmHg) from baseline in both groups- controls 130.3±21 to 128.3±15; intervention 130.3±21 to 127.6±15 (p<0.01 for before and after comparison) but there was no difference between the two groups at 12 months (p=0.18). |
| Kaaya SF, et al. [92] 2013 | Multiple: Mental health; Other: HIV | Rehabilitation | Nurse/social worker/midwife: provide problem-solving therapy, psychosocial education and therapy, group counseling, education on safer sexual practices with the standard of care. | Nurse, social workers, Midwife | Depressive symptoms and increasing prenatal disclosure rates of HIV status | 60% of women in the intervention group were depressed post-intervention, versus 73% in the control group [Relative Risk (RR)-0.82, 95% confidence interval (CI): 0.671.01,p-0.066]. HIV disclosure rates did not differ across the two study arms. |
| Kalani Z, et al. [27] 2022 | Other: stroke | Prevention; promotion; Referral | Assessment of the probability of pneumonia, urinary tract infection. Planning: individualised care plan Educating the patient and family and follow up | Nurse | Incidence of pneumonia and urinary tract infection | No difference in the incidence of pneumonia between the two groups (11.6% vs. 19.2%,P= 0.35). Significant difference in the incidence of urinary tract infection (0% vs. 24.6%,P< 0.001) |
| Kara M, et al. [140] 2007 | Respiratory: COPD | Promotion; Management | Nurse: education of the patient, provision of continuity of care | Nurse | Outcomes of patients with COPD | Statistically significant decrease in nursing diagnoses in favour of the experimental group. |

(Continued)

**Table 2.** (Continued)

| Author, Year | Disease | Activity | Management details for MLHP | Workforce | Primary outcome | Primary outcome result |
|---|---|---|---|---|---|---|
| Kargar JM et al.[85] 2015 | Mental health | Management; Promotion | Telephone consultation, Psychosocial education and therapy. | Nurse | Depression, anxiety and stress scores | Significant differences were observed between the two groups in the post-test regarding the dimensions scores of DASS scale |
| Keliat B, et al.[141] 2020 | Mental health | Management | Early detection, stress management, counselling therapy, family empowerment. | Nurse | Aspects of function-ing ("life skill") | Significant difference in scores before and after the implementation in the intervention group (19.94 ± 1.27 and 38.83 ± 9.32) with p <.001 and the control group (26.93 ± 12.50 and 30.89 ± 12.41) with p =.002. |
| Khetan A et al.[142] 2019 | CVD: Hypertension | Prevention; Promotion; Screening; Management | Counselling to people with hypertension and diabetes. Encouraging physician visits, medication purchase, and medication adherence. | CHW | Blood pressure | The mean ± SD change in SPB at 2 years was −12.2 ± 19.5 mm Hg in the intervention group as compared with −6.4 ± 26.1 mm Hg in the control group. (p = 0.001). |
| Khezri E, et al. 2022[26] | Cancer: Breast cancer | Rehabilitation | Deliver spiritual support sessions | Nurse | Scores of hope | Mean scores of hope for the intervention and control groups were 46.71 and 40.40, respectively (P<0.05). |
| Kondal D, et al.[143] 2022 | CVD: Hypertension | Prevention | Custom-made lifestyle modification | ASHAs | Mean change in SBP | The mean SBP increased in both the intervention and control groups at 18- months post intervention. The mean SBP increased from 124.4 mm Hg to 126.7 mm Hg and 125.7 to 126.1 mm Hg in the intervention and control group respectively. The cluster adjusted mean SBP difference was 1.91 mm Hg (95% confidence interval: -0.02 to 3.85). |
| Labhardt ND, et al.[93] 2011 | Multiple: Hypertension; Diabetes | Management; Promotion | Counselling, understanding obstacles in long-term care and adherence support for patients | Nurse | Patient retention | Retention rates in the incentive and letter group were 60% and 65%, respectively, compared with 29% in the control group. The differences between the groups were significant (95% CI: 21% to 46%;P< 0.001). |
| Li P, et al. [144], 2015 | Respiratory: COPD | Management; Rehabilitation | Education and guidance about use of inhalers, supporting their care plan including exercises for rehabilitation. | Nurse | Preventing acute exacerbations, improving health-related quality of life among patient with COPD | Total and subscale scores (P < 0.05) of SOLDQ and CSES significantly improved compared to the baseline ones in the intervention group. At 12 weeks, scores showed a sustained and significant growth in the intervention group (P < 0.05). Significantly lower average medical expenses than the control group (P < 0.05) |

*(Continued)*

**Table 2.** (Continued)

| Author, Year | Disease | Activity | Management details for MLHP | Workforce | Primary outcome | Primary outcome result |
|---|---|---|---|---|---|---|
| Li X, et al.[47], 2020 | Respiratory: COPD | Promotion, Management | Education about COPD, diet and physical activity and follow-up with adherence support (for inhalers) | Nurse | Quality of life | Significant differences in the total quality of life score (20.29±10.03 vs 30.14±12.52) between the intervention group and the control group (P<.05). |
| Li C, et al.[145] 2023 | Cancer: Gastrointestinal cancer | Prevention and Screening | Nurse-led counselling, health education, tailored information, addressing hepatocellular carcinoma screening barriers, | Nurse | Hepatocellular carcinoma screening rates after 6-months | Significant differences in the hepatocellular carcinoma screening uptake rate (75.6%vs.42.1%;χ2=17.909;p<0.001) in the intervention group compared to the control group. |
| Liang R, et al.[146] 2012 | Diabetes | Prevention; Promotion; Management | Education on foot care, foot examination, follow-up visits | Nurse | Knowledge and foot care behaviours | At 1 and 2 years later, diabetes knowledge and foot care behaviour was significantly higher in the intervention than control group (p<0.05 or p<0.01) |
| Liu CY, et al.[147] 2010 | Cancer: Breast cancer | Promotion; Screening | Deliver education for breast cancer examination | Nurse | Self-examination frequency | 34% of the intervention, 11% of the control, group did breast self-examination (P<0.001). |
| Liu H, et al.[35] 2015 | Diabetes | Management; Promotion | Education and management of diet | Dietitian | Anthropometric measurements | No significant change in anthropometric measures |
| Lund C, et al.[68] 2020 | Mental health | Management | Counselling, psychoeducation, problem-solving skills, behavioural activation, healthy thinking, relaxation training, social support, screening. | CHW | Reduction in perinatal depression | There were no differences between the two arms at eight months gestation, three months, or 12 months postpartum. |
| Ma C, et al.[148] 2014 | CVD: Hypertension | Management | Counselling and adherence support | Nurse | Blood pressure | SBP, DBP of the intervention group decreased compared with ones of the control group, the difference values were 4.92 and 2.58, respectively. |
| Malakouti SK, et al. [86], 2016 | Mental Health | Management | Consultation, referral, review of treatment. | Nurse | Hospitalization rate | Rate of rehospitalization for the telephone follow-up and as-usual groups were respectively 1.5 and 2.5 times higher than the home-visit group |
| Mash RJ, et al.[149] 2014 | Diabetes | Prevention | Education on disease and medication, lifestyle education | CHW | Primary outcomes were diabetes self-care activities, 5% weight loss and a 1%reduction in HbA1clevels | No significant improvement was found in outcomes, apart from a significant reduction in mean systolic (-4.65 mmHg, 95% CI 9.18 to -0.12; P=0.04) and diastolic blood pressure (-3.30 mmHg, 95% CI -5.35to -1.26; P=0.002) |

*(Continued)*

**Table 2.** (Continued)

| Author, Year | Disease | Activity | Management details for MLHP | Workforce | Primary outcome | Primary outcome result |
|---|---|---|---|---|---|---|
| Mehralian H, et al. [150], 2014 | CVD: Heart Failure | Promotion | Monitoring patient progress and outcomes | Nurse | Quality of life | At hospital discharge, mean score of all 8 sub-score of SF-36 questionnaire was 63.4 in patients of intervention and 61.1 in patients of control, respectively (P> 0.05). After 6 months, mean score of QOL was higher in intervention than in control. |
| Mini GK, et al. [25], 2022 | CVD: Hypertension | Management; Promotion | Educate schoolteachers about hypertension control, healthy lifestyle practices, and self-management of hypertension and related NCDs and risk factors. | Nurse | Blood pressure | A greater proportion of intervention participants (49.0%) achieved hypertension control than the usual care participants (38.2%) |
| Mittra I, et al. [151] 2021 | Breast cancer | Prevention; Screening | Deliver educational intervention | CHW | Downstaging of breast cancer at diagnosis and reduction in mortality from breast cancer | Breast cancer was detected at an earlier age in the intervention group than in the control group (age 55.18 (standard deviation 9.10) v 56.50 (9.10); P=0.01), with a significant reduction in the proportion of women with stage III or IV disease (37% (n=220) v 47% (n=271), P=0.001). |
| Mollaoğlu M, et al. [18] 2009 | Diabetes | Prevention | Dietary education, exercise, measuring of blood and urine glucose | Nurse | HbA1c | Intervention arm mean HbA1c values fell from 9.5 Â± 1.7 mg/dl to 7.5 Â± 1.3 mg/dl. Control group, decreased from 9.7 Â± 1.6 mg/dl to 9.6 Â± 1.6 mg/dl. The difference between the groups was statistically significant (p <0.05) |
| Moreira R, et al. [152] 2015 | Diabetes | Management; Promotion | Create individualised care plan, Blood pressure and glucose monitoring, evaluate insulin application technique, assess proper disposal of contaminated materials, promote health education (capillary glycemia test, insulin administration, foot care). | Nurse | HbA1c | HbA1c was reduced from an average of 10.33% to 9.0% (p <.01) in the intervention group and from 9.57% to 8.93% (p =.05) in the control group; the group by time effect was not significant. |
| Muchiri JW, et al. [36] 2016 | Diabetes | Management | education programme (health diet advice, meal planning, meal preparation), understanding diabetes | Dietitian | HbA1c | Differences in HbA1c (primary outcome) were -0·64% (P=0·15) at 6 months and -0·63% (P=0·16) at 12 months in favour of the intervention group |

*(Continued)*

**Table 2.** (Continued)

| Author, Year | Disease | Activity | Management details for MLHP | Workforce | Primary outcome | Primary outcome result |
|---|---|---|---|---|---|---|
| Myers B, et al.[153] 2022 | Mental health | Management | Health promotion and adherence support for HIV and non-communicable diseases in general, mental health education, use motivational interviewing techniques to build rapport and develop readiness to change, develop a change plan | CHW | Changes in depression and alcohol use | The trial had three groups and there were significant differences in the depression scales at 6 months (-4.96; 95% confidence interval: -7.47 to -2.45; p<0.0001) between the dedicated and treatment as usual group and between the designated and treatment as usual group (-2.76; 95% confidence interval: -5.28 to -0.26; p=0.031). After 12-months the scores were also statistically different; (-5.02; 95% confidence interval: -7.51 to -2.54; p<0.0001) between the dedicated and treatment as usual group and between the designated and treatment as usual group (-6.38; 95% confidence interval: -8.89 to -3.88; p<0.0001). On Alcohol Use Disorders Identification Test (AUDIT); at 6 months there were statistically significant differences (-2.75; 95% confidence interval: -5.31 to -0.19; p=0.035) between the dedicated and treatment as usual group and between the designated and treatment as usual group (-2.80; 95% confidence interval: -5.41 to -0.19; p=0.035). After 12-months the scores were not statistically different; (-1.73; 95% confidence interval: -4.38 to 0.92; p=0.20) between the dedicated and treatment as usual group and between the designated and treatment as usual group (-1.97; 95% confidence interval: -4.65 to 0.71; p=0.15). |
| Neupane D, et al.[154] 2018 | CVD: Hypertension | Prevention; Management; Referral | Lifestyle counselling and blood pressure monitoring | CHW | Blood pressure | Mean SBP at 1 year was significantly lower in the intervention group than in the control group for all cohorts: the difference was -2·28 mm Hg (95% CI -3·77 to -0·79, p=0·003) for participants who were normotensive, -3·08 mm Hg (-5·58 to -0·59, p=0·015) for participants who were prehypertensive, and -4·90 mm Hg (-7·78 to -2·00, p=0·001) for participants who were hypertensive |

*(Continued)*

**Table 2.** (Continued)

| Author, Year | Disease | Activity | Management details for MLHP | Workforce | Primary outcome | Primary outcome result |
|---|---|---|---|---|---|---|
| Ogedegbe G, et al. [155, 156] 2018 | CVD: Hypertension | Promotion; Management | Cardiovascular risk assessment, lifestyle counselling, and initiation/titration of antihypertensive medications | Nurse | Blood pressure | Across both groups (healthcare insurance only and healthcare insurance + nurse visit), there was a significant and sustained reduction in SBP. |
| Okube OT, et al.[157] 2023 | CVD: CVD risk | Health promotion and prevention | Individualised and group-based health education at three time points, education on major CVD behavioural risk factors, including advice on salt and sugar reduction, avoidance or reduction on processed/fast foods and saturated fats. Dietary advice on plating and correct portion sizes. | Nurse | Changes in CVD knowledge level | Significant differences in the CVD knowledge levels from baseline (74.4% vs 29.0%; p<0.0001) in the intervention group compared to the control group. |
| Osborn TL, et al. [158], 2021 | Mental health | Prevention; Promotion; Management | Layperson-delivered group intervention on growth mindset, gratitude, and value affirmation | Lay-person | Depression and anxiety symptoms | Reductions in depressive symptoms at post-treatment (Cohen d= 0.35 [95% CI, 0.09-0.60]), and 7-month follow-up (Cohend= 0.45 [95% CI, 0.19-0.71]) Reductions in anxiety symptoms at post treatment (Cohen d= 0.37 [95% CI,0.11-0.63]), and 7-monthfollow-up (Cohen d= 0.44 [95% CI, 0.18-0.71]). |
| Pace L, et al.[159] 2018 | Cancer: Breast cancer | Screening; Promotion; Referral | Nurse: provide screening breast concerns, and refer patients to the breast clinic<br>CHW: provide educational sessions and encourage individual women with breast symptoms to seek prompt evaluation at health centers | Nurse and CHWs | Health care use and patient outcomes | 1,500 patients sought care at intervention HCs for breast concerns versus 600 at control HCs. Biopsy rate was 36.6 per 100,000 person-years from intervention versus 8.9 per 100,000 from control arm (P <.001). |
| Pan Y et al. [160] 2019 | Mental health | Prevention; Promotion; Management | Nurse: cognitive behavioural interventions, consultation | Nurse | Depressive symptoms and coping strategies | Significant difference in depressive symptoms and active coping between groups over time, with p<.001 for the interaction between depressive symptoms and groups and p<.01 for the interaction between active coping and groups. |
| Pan Y, et al.[202] 2019 | Mental health | Promotion; Rehabilitation | Psycho-education, Cognitive behavioural therapy | Nurse | Depressive symptoms of caregivers of persons | Improved mutuality (p = 0.049) and active coping (p = 0.0001) and decreased passive coping (p = 0.001) were found to predict the reduction of depressive symptoms among caregiver. |
| Patel V, et al.[43, 161] 2010 | Mental health | Management; Promotion; Referral | Lay health counsellor: deliver psychoeducation, interpersonal psychotherapy, treatment prescription and administration<br>Physician: support and supervise the lay health counsellor, follow, and refer high risk participants<br>Psychiatrist: assess high-risk participants and consult a primary health physician | Lay health counsellors, primary care physician and a psychiatrist (clinical specialist) | Recovery from common medical disorders | The intervention had no significant effect on recovery from common mental disorders in patients with depression. |

*(Continued)*

**Table 2.** (Continued)

| Author, Year | Disease | Activity | Management details for MLHP | Workforce | Primary outcome | Primary outcome result |
|---|---|---|---|---|---|---|
| Peiris D, et al. [54] 2019 | CVD: Hypertension | Screening; Management; Referral | CHWs: screening, education, referral, follow-up, adherence support Doctor: diagnoses, initiation of treatment | CHW, Doctor | Blood pressure | No difference in the proportion of people achieving SBP targets (41.2% vs 39.2%; adjusted odds ratio (OR) 1.01 95% CI 0.76–1.35) or receiving BP-lowering medications in the intervention vs control periods respectively. |
| Petersen I, et al.[162] 2021 | Mental health | Management; Referral; Promotion | Counselling, treatment prescription, referral | Nurse | Depressive symptoms | Retention was good and participants in the group based IPT intervention showed significant reduction in depressive symptoms on completion |
| Pisani P, et al.[163] 2006 | Cancer: Breast cancer | Prevention; Promotion; Referral | Nurse and midwives: provide mass screening by clinical examination of the breast (CBE), educate technique of breast self-examination, and refer women with positive results. Doctor: perform needle biopsies | Nurse, Midwife, Doctor | Incidence of breast cancers | In the single screening round 92% accepted, 3,479 were detected and referred for diagnosis. Of these 35% completed diagnostic follow-up |
| Prabhakaran D, et al. [53] 2018 | Multiple: Hypertension; Diabetes; Mental health | Management | Lifestyle education, Treatment prescription, Review treatment, consultation, blood pressure and glucose measurement. | CHW | Blood pressure and HbA1c | No difference in change in SBP and HbA1c between the 2 arms. |
| Pradeep J, et al.[164] 2014 | Mental health | Prevention; Promotion; Referral; Management | Screening, Pill count, adherence counselling, referral, Education about disease, Monitoring side effects, prescription | CHW | Treatment seeking and completion Changes in severity of depression and changes in quality of life | Significantly greater number of women from the intervention group completed the treatment and were on treatment for a longer duration compared to the control group. No significant differences in the severity of depression or quality of life between the groups. |
| Rahman A, et al. [165] 2008 | Mental health | Promotion; Management | Consultation, Counselling, referral | Nurse | Infant nutrition | The differences in weight-for-age and height-for-age Z scores for infants in the two groups were not significant at 6 or 12 months |
| Rahman A, et al.[166] 2016 | Mental health | Management; Rehabilitation; Promotion | Problem solving therapy, behavioural activation, strengthening social support, stress management, adherence counselling | Nurse | Anxiety and depression symptoms | After 3 months of treatment, the intervention group had significantly lower mean (SD) HADS scores than the control group for anxiety (7.25 [3.63] vs 10.03[3.87]) and depression (6.30 [3.40] vs 9.27 [3.56]) |
| Rahul A, et al.[167] 2021 | Diabetes | Management | Lifestyle education (dietary advice, encourage physical activity, cessation of smoking and alcoholism, drug compliance), treatment adherence, understanding the disease, monitoring glucose | CHW | Fasting blood sugar | mean fasting blood sugar values dropped in both groups, the drop being higher in the intervention group. The intervention was associated with a significant reduction in FBS at the end of 6-month follow-up after controlling for the effect of baseline (p<0.001) |

*(Continued)*

**Table 2.** (Continued)

| Author, Year | Disease | Activity | Management details for MLHP | Workforce | Primary outcome | Primary outcome result |
|---|---|---|---|---|---|---|
| Rylance S, et al.[168] 2021 | Respiratory: COPD | Management | Nurse: Clinical assessment, optimisation of inhaled treatment Lay Educators: individualised asthma education | Lay educators, nurse | Asthma symptom control | At 3 months, children in the intervention arm had a mean (SD) cACT of 22.9 (2.3), compared with 20.8 (3.0) in stan-dard care (p<0.001). Children receiving the intervention had a greater mean (SD) change in cACT score from baseline; 2.7 (2.8) compared with 0.6 (2.8) for standard care participants, a differ-ence of 2.1 points (95%CI: 1.1 to 3.1, p<0.001) |
| Saffi M, et al.[169] 2014 | CVD: CVD risk | Prevention; Management | Education and follow-up Develop exercise and dietary goals with patients | Nurse | 10-year cardiovascu-lar risk scores | A 1.7 point (à°13.6%) reduction in risk score was recorded in the inter-vention group, vs a 1.2 point increase in risk score (+11%) in the control group (p=0.011 |
| Safren SA, et al.[170] 2021 | Mental health | Promotion; Management; Rehabilitation | Cognitive Behavioural Therapy: adherence counselling, psychoeducation, mood monitoring, problem solving, Relaxation training, relapse prevention | Nurse | Depression and medi-cation adherence | t 4 months, the HAMD scores in the CBT-AD condition improved by an estimated 4.88 points more, and for weekly adherence, 1.61 percentage points more per week than ETAU. |
| Salimzadeh H, et al.[172,173] 2018 | Cancer: Gas-trointestinal cancer | Prevention; Screening | Nurse: provide motivational interview counsel-ling sessions and training | Nurse | Colonoscopy in first degree relatives within six months | At follow-up, the uptake of screening colonoscopy in the intervention group was 83.5% versus 48.2% in controls (P<.001). |
| Saisanan Na Ayudhaya W, et al.[171] 2020 | Mental health | Management | CHW: Screening Nurse: behaviour activation aimed to increase engagement in activities and follow up patients | Nurse, CHW | Thai Geriatric Depression Scale and Depression Anxiety Stress Scales | The adjusted mean change in depres-sion scores improved significantly in the intervention group compared to the usual care-only group, and anxiety score improved significantly at 6 months |
| Samonnan T, et al.[174] 2018 | Mental health | Promotion; Rehabilitation | Researcher: serve as interventionist and guide on the intervention for participants to rais-ing self-awareness, Support self-care, lifestyle education Nurse: provide routine care, including health-related education about chemotherapy | Nurse, researcher | Psychological symp-tom experiences | Non-significant difference in the mean scores of psychological symp-tom experiences between the two groups, but there was a significant time difference and a significant interaction effect. |
| Sankararnayanan R, et al.[175] 2007 | Cancer: Cervi-cal cancer | Screening; Promotion; Referral | Identifying women eligible for screening. Collecting and managing data on population screening. Doctors: supervising nurses, performing LEEP | CHW | Incidence and mortality | The intervention group had lower cervical cancer incidence and mortality rates than the control group. Overall, the intervention group had a significant reduction in cervical cancer incidence and a significant reduction in cervical cancer mortality compared with the control group. |

*(Continued)*

**Table 2.** (Continued)

| Author, Year | Disease | Activity | Management details for MLHP | Workforce | Primary outcome | Primary outcome result |
|---|---|---|---|---|---|---|
| Sartorelli D, et al.[37] 2005 | CVD: CVD risk | Prevention | Individualised nutritional counselling sessions | Dietitian | Cholesterol | At 6 months total cholesterol (−12.3% vs.−0.2%), low-density lipoprotein (LDL) cholesterol −15.5% vs.+4.0%) and fasting plasma glucose (−5.6% vs.+0.1%) in the intervention group compared with the control group(P<0.05). |
| Scain SF, et al.[176] 2009 | Diabetes | Promotion, Management | Lifestyle education (self-care, healthier dietary habits, exercise, self-monitoring, foot care), understanding of diabetes and its complications. | Nurse | HbA1c | HbA1C levels decreased significantly in the intervention group after the 4th month and remained lower than in the control group until the 12th month. |
| Scazufca M, et al.[177],2022 | Mental Health | Prevention and Management | The psychosocial intervention consisted of a 17-week programme delivered by CHWs. Through these sessions, they monitored depression symptoms, checked chronic conditions and antidepressant medication adherence (if on medication), psychoeducation about depression, simple strategies to improve mood, discuss difficult cases with the team, Behavioural activation (introduce pleasant activities in a stepped way), introduce relapse prevention strategies, inform team members about the end of the intervention and summarise case. | CHW | Depression recovery at 8-months | Significant differences in the recovery from depression at 8-months (62.5%vs.44.0%; OR=2.16 (95% confidence interval: 1.47–3.18); p<0.0001) in the intervention group compared to the control group. |
| Schwalm JD, et al.[29] 2019 | CVD: CVD risk | Promotion, Screening; Management | Screening, health education, adherence support, recommended treatment, in some cases they delivered medicines to patients Doctor: prescribed medicines | Nurse | CVD risk | Statistically significant reduction in Framingham Risk Score for 10-year CVD risk |
| Segginli S, et al.[178] 2011 | Cancer: Breast cancer | Promotion, Screening | Developing and distributing educational materials. | Nurse | Screening | The breast health promotion program significantly increased breast self-examination frequency. No significant differences existed in mammography and clinical breast examination rates between the two groups at 6 months. |
| Selvaraj F, et al.[179] 2012 | CVD: CVD risk | Management; Promotion | Provided self-management support and patient empowerment, reinforcement of the health education information and adherence support | Nurse | LDL-C | LDL-C were 30.09% and 27.54% for the intervention group and control groups, respectively. The difference of mean change in the intervention group was 2.55% lower than the control group(p=0.288). |
| Sharma KK, et al. [180], 2016 | CVD: Other heart disease | Promotion, Management | Follow-up care | CHW | Medication adherence | At 12 and 24 months, respectively, in intervention vs control groups adherence (>80%) |

*(Continued)*

**Table 2.** (Continued)

| Author, Year | Disease | Activity | Management details for MLHP | Workforce | Primary outcome | Primary outcome result |
|---|---|---|---|---|---|---|
| Shastri SS, et al.[20] 2014 | Cancer: Cervical cancer | Promotion, Screening | Identifying women eligible for screening. Collecting and managing data on population screening. Doctors: supervising PHWs Community rapport-building procedures | CHW | Mortality. | The screening group showed a statistically significant 31% reduction in cervical cancer mortality (RR = 0.69; 95% CI = 0.54 to 0.88; P =.003) |
| Shelley D, et al.[181] 2021 | Other: Tobacco use | Prevention, Promotion, Management | PHC workers: Used the 4As to educate, support and manage patients: Ask about tobacco use, Advise to quit, Assess readiness, Assist with brief counselling CHW: intensive counselling for tobacco cessation | PHC providers, CHWs | Adoption of the 4As or 4As + intensive counselling | Adoption of the 4As increased significantly across both study arms (all p <.001). |
| Shi Y, et al.[182] 2020 | Multiple: Cancer: Cervical cancer; Mental health | Promotion, Rehabilitation | Referral, Sexual psychological rehabilitation, counselling, sexual yoga and pelvic floor rehabilitation exercises, Sex education | Nurse | Sexual Function | Compared with participants in the control group, participants in the intervention group showed significant improvements in sexual function and improvements in their levels of depression and well-being |
| Siabani S, et al.[183] 2016 | CVD: Heart Failure | Management, Promotion | Community health volunteer: provide home-based face-to-face self-care education for patients with chronic heart failure Nurse and a general practitioner: provide education in the hospital | Community health volunteers(CHV), Nurse, General practitioner | Self-care components | After intervention, self-care components were significantly increased in all three groups, with the exception of self-care confidence, which did not change significantly in the usual care group. Self-care scales in patients educated by CHVs improved to a similar extent to patients educated by nurse and general practitioners. |
| Sinha B, et al.[184] 2021 | Mental health | Prevention; Promotion | Consultation, breastfeeding counselling, referral | Nurse | Depressive symptoms | The proportion of mothers with moderate-to-severe postpartum depressive symptoms was 10.8% (95% CI, 8.9%-12.9%; 105 of 974 mothers) in the intervention group vs 13.6% (95% CI, 11.4%-16.1%; 116 of 852 mothers) in the control group |
| Steffen PLS et al.[185] 2021 | Multiple: Hypertension; Diabetes | Management | Motivational interviewing | Nurse | HbA1c | 0.4% (p<0.01) reduction in HbA1c levels for the intervention group with a statistical significance and a small effect size (0.3) |
| Subramanian SC, et al.[186] 2020 | Diabetes | Management; Promotion | Prescription, teaching on diabetes and its complications, education on treatment adherence, lifestyle education (self-management, exercise, healthier dietary habits) | Nurse | Diabetes Self-Management Questionnaire | Significant increase in physical activity in intervention (2.17 to 9.02) and decrease in control (1.28 to 1.22). No significant differences in dietary control, blood glucose monitoring, medication adherence, physician contact |

*(Continued)*

**Table 2.** (Continued)

| Author, Year | Disease | Activity | Management details for MLHP | Workforce | Primary outcome | Primary outcome result |
|---|---|---|---|---|---|---|
| Sun J, et al.[187] 2008 | Diabetes | Prevention; Management | Education on self-management, Lifestyle education (behavioural and lifestyle modifications including physical activity and healthier dietary habits) | Dietitian | Fasting blood glucose, insulin, systolic and diastolic blood pressures | The Intervention Group improved fasting blood glucose, insulin, systolic and diastolic blood pressures compared to Reference Group (p<0.05). |
| Sun Y, et al.[22] 2022 | CVD: Hypertension | Management | Treatment initiation, ensure appropriate dosage for each patient, health coaching on home blood pressure monitoring, medication adherence and lifestyle changes. | CHW | Blood pressure <130/80 mm Hg at 18 months | There were statistically differences in blood pressure changes between the intervention and control group at 18-months (group difference=37.0%; 95% confidence interval=34.9 to 39.1%; p<0.0001). In the intervention group, 57.0% of patients had a blood pressure of less than 130/80 mm Hg compared to 19.9% in the control group. |
| Temucin E, et al.[188] 2018 | Cancer: Gastrointestinal cancer | Prevention; Promotion; Screening | Screening, navigation and counselling | Nurse | Screening participation | The FOBT (82 and 84%, respectively) and colonoscopy completion rates (15 and 22%, respectively) were significantly higher in the intervention versus control groups at 3 and 6 months follow-up. |
| Thakur D, et al.[189] 2019 | Multiple: Mental health; Other: intracranial tumours | Management; Rehabilitation; Promotion | Assessment of behavioural symptoms, counselling, diagnosis, consultation, discharge education, prevention of complications | CHW | Behavioural symptoms | Patients in the intervention group had significantly fewer behavioural symptoms and less severity of behavioural symptoms as compared to the control group. |
| Tian M, et al.[33] 2015 | CVD: Hypertension | Promotion; Management | China: CHWs were aided by the smartphone-based app to prescribe medication and lifestyle counselling. India: CHWs screened and provided counselling to patients using an app. Doctors: prescribed the medications | CHW | Adherence to anti-hypertensive medication | The intervention group had a 25.5% (P<0.001) higher net increase in the proportion of patient-reported anti-hypertensive medication use pre- and post-intervention. |
| Tomlinson M, et al.[190] 2018 | Mental health | Prevention; Management; Promotion | Education on HIV disclosure, alcohol prevention education, breastfeeding education and counselling | CHW | Physical and cognitive outcomes | No significant differences |
| Vedanthan R, et al.[51] 2019 | CVD: Hypertension | Screening; Management; Referral; Promotion | 1 group of CHWs used paper-based tools to provide tailored behavioural communication; Another group of CHWs used smartphones to provide tailored behavioural communication | CHW | Impact on linkage to care | Linkage to care was 49% overall, with significantly greater linkage in the usual care (50%) and smartphone arm (54%) than paper-based arm (43%) |
| Wagner G et al.[191] 2016 | Multiple: Mental health; Other: HIV | Management | Diagnosis of depression; initiating antidepressant therapy (using a protocol/algorithm) | Nurse | Screening | Mean number of adult clients seen each clinic day across all sites in the structured protocol arm was 69.1 (SD = 33.5; range: 35.6-97.8 across sites), which was more than the 58.0(SD = 30.6; range: 29.4-90.3) clients seen in the clinical acumen arm (t = 7.35, df = 1641, p<.0001). |

*(Continued)*

**Table 2.** (Continued)

| Author, Year | Disease | Activity | Management details for MLHP | Workforce | Primary outcome | Primary outcome result |
|---|---|---|---|---|---|---|
| Wang Y, et al.[192] 2014 | Respiratory: COPD | Management; Promotion | Assist patients to understand the susceptibility and severity of COPD, understand benefits of treatment, Education for healthy behaviours, monitor their symptoms. | Nurse | Health belief and self-efficacy, degree of difficulty in breathing, activities of daily life, six-minute walking distance and pulmonary function indicators. | Patients in the intervention group had significantly higher mean total scores in the Health Belief Scale and the COPD Self-Efficacy Scale, and in all the sub scales, than in the control group except the perceived disease seriousness. Results showed that the value of FEV1/FVC ratio had a significant difference between study groups before and after the intervention. |
| Wang LH, et al.[193] 2020 | Respiratory: COPD | Promotion; Management; Prevention | Patient assessment for physical activity self-management needs, management of COPD, follow up of patients | Nurse | COPD-related hospital admissions and emergency department visits | Compared to the control group, participants in the intervention group showed significantly fewer COPD-related hospital admissions (P = 0.03) and emergency department visits (P = 0.001) |
| Wang G, et al.[194] 2021 | Cancer: Breast cancer | Management; Promotion | Education about the condition, dealing with relationships and planning for the future | Nurse | Post traumatic growth, anxiety and depression | Compared to control group, participants in intervention group reported higher level of post-traumatic growth (p < 0.01 or 0.05), reduced anxiety and depression (p < 0.01 or 0.05 and p < 0.01 or 0.05). |
| Weiss WM, et al.[195] 2015 | Mental health | Management; Rehabilitation; Promotion | Counselling, Cognitive processing therapy, psychoeducation, Referral, | CHW | The Posttraumatic Growth Inventory (PTGI) | Significant improvements on PTG were observed in the intervention group compared with the control group |
| Wroe EB, et al.[196] 2021 | Hypertension; Diabetes; Asthma; Mental health, malnutrition and TB, family planning, antenatal care | Prevention; Screening; Management; Referral | Lifestyle education, screening, Referral, Treatment adherence monitoring | CHW | Proportion of NCD clients who default from care this month | Decrease of approximately 20% in the rate of patients defaulting from chronic NCD care each month (−0.8 percentage points (pp) (95% credible interval: −2.5 to 0.5)) while maintaining the already low default rates for HIV patients (0.0 pp, 95% CI: −0.6 to 0.5). |
| Xavier D, et al.[197] 2016 | CVD: CVD risk | Prevention; Management | Educate patients on healthy lifestyle and drugs, and measures to enhance adherence. | CHW | Adherence to medications | Adherence (≥80%) to prescribed evidence-based drugs was higher in the intervention group than in the control group (97% vs 92%, odds ratio [OR] 2·62, 95% CI 1·32–5·19; p=0·006) |
| Xu DR, et al.[52] 2019 | Mental health | Management; Promotion; Referral | Adherence support, Observed treatment ingestion. | Lay health worker | Medication adherence | Medication adherence was 27% greater in the intervention group (0.61) than in the control group (0.48). |

*(Continued)*

**Table 2.** (Continued)

| Author, Year | Disease | Activity | Management details for MLHP | Workforce | Primary outcome | Primary outcome result |
|---|---|---|---|---|---|---|
| Xueyu L, et al.[198] 2015 | CVD: CVD risk | Rehabilitation; Management; Promotion | Nurse led rehabilitation training | Nurse | Quality of life | No significant direct effects for group for SF-36. The intervention group showed improvement in physical functioning role-physical, bodily pain, and vitality (p < 0.05) on the SF-36. |
| Yan H, et al.[44] 2021 | CVD: CVD risk | Management; Promotion | Nurse: provide usual care and nurse-led multi-disciplinary team management | Nurse | Cardiovascular hospitalization and cardiovascular death. | Patients under intervention (a nurse-led multidisciplinary team approach) showed fewer cardiovascular hospitalizations (17 vs. 35,p=0.006) than those receiving usual care. No difference was found with cardiovas-cular death. |
| Yin Z, et al.[199] 2018 | Diabetes | Management | Goal setting and education (diet, physical activ-ity) of patients | Nurse | Diabetes | No significant reduction in blood glucose |
| You J, et al.[200] 2020 | CVD: Heart Failure | Promotion; Management | Monitoring patient progress and outcomes | Nurse | Medication adherence | No significant differences |
| Yuan X, et al.[201] 2015 | Respiratory: COPD | Prevention; Management | Health education, smoking cessation counselling, and education on management of COPD | Nurse | Incidence, decline in lung function, and mortality | COPD incidence was lower in the intervention group than in the control group (10%vs16%, P<0.05). Intervention group had a signifi-cantly lower cumulative COPD-related death rate than the control group (37%vs47%, P<0.05). |
| Zhang P, et al.[203] 2017 | CVD: Isch-aemic heart disease | Management; Promotion | Structured assessment and health education, health education and coaching via home visits and phone calls | Nurse | BP, fasting blood glucose, triglycerides, high density and low-density lipopro-tein cholesterol | Clinical outcomes showed significant differences between the control and intervention groups over time. |
| Zheng X, et al.[204] 2020 | CVD: CVD risk | Promotion; Management | Face to face education sessions with follow up via phone calls. | Nurse | Cardiovascular risk | Decreased cardiovascular risk was found in the intervention group, but no significant group-by-time effect was detected |
| Zhu X, et al.[48] 2018 | CVD: Hypertension | Management; Referral; Promotion | Nurse: home visits, telephone follow-ups and referral when necessary. Doctor: pharmacological treatment for referred patients. | Nurse, doctor | Blood pressure | SBP reduction was about 14.72mmHg in the intervention, which was greater than that in the control group(9.22mmHg). DBP decreased more in the intervention (7.43mmHg) than in the control group (5.14mmHg). |

Sd=standard deviation; SBP=systolic blood pressure; RR=Risk Ratio, Cardiovascular disease.

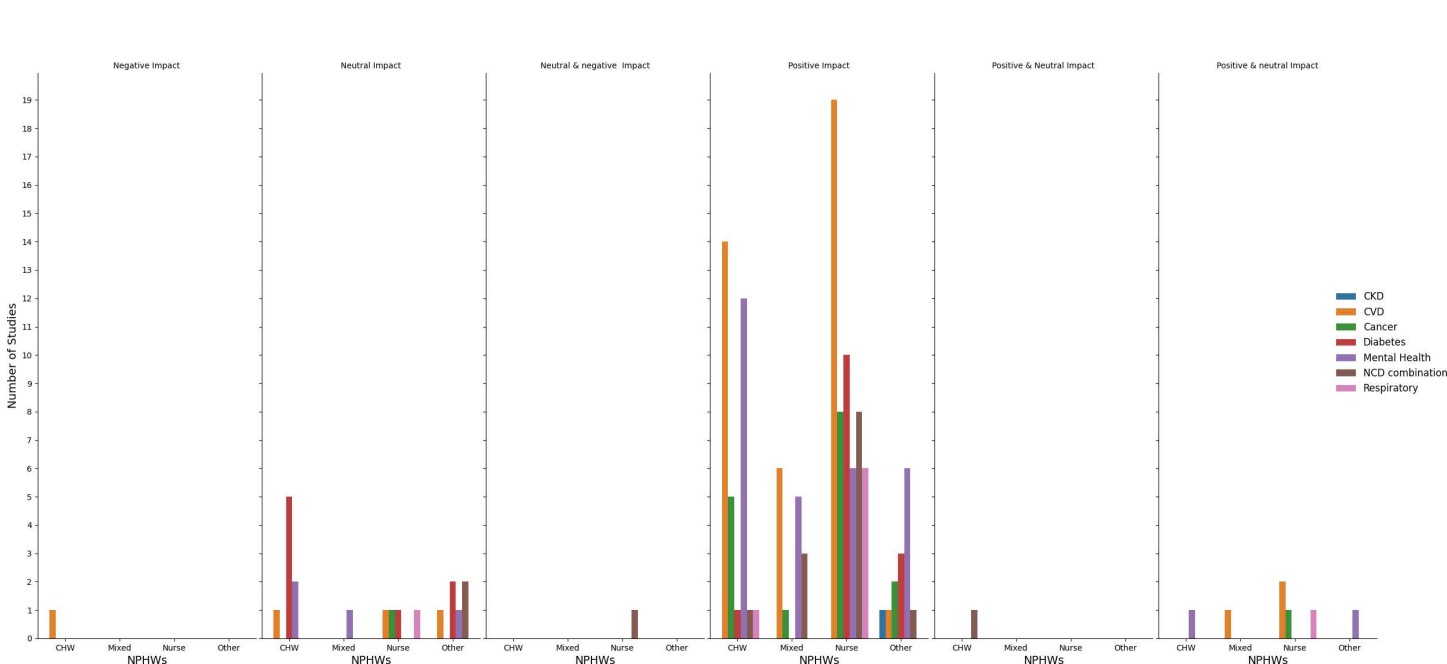

**Fig 3. Number of studies by NCD conditions, types of NPHWs, and primary outcomes.** Footnote. **Nurse**: Includes all roles involving nurses, midwives, and advanced practice nurses. **CHW**: Refers to community health workers, lay health workers, and similar roles. **Mixed**: Combines roles, such as CHWs and nurses, or interdisciplinary teams, including physicians. **Other**: Encompasses roles that do not fall into the above categories, such as dietitians and researchers.

studies where the primary outcome was not achieved, results demonstrated that task-sharing for NCD prevention and control was acceptable, feasible, and resulted in better treatment uptake[53,60–62].

Moreover, Fig 3 illustrates the distribution of studies across various NCD conditions, types of NPHWs, and their associated primary outcomes. A significant number of studies reported positive outcomes for CVD when CHWs and nurses were involved. Additionally, larger studies highlighted positive outcomes for mental health conditions with CHWs, while nurses showed positive results for diabetes management.

## What does process evaluation reveal about task-sharing?

The enablers of these studies included the context in which NPHWs carry out their activities, their relationship and trust with the community, support by the leadership and training provided[52,61,63,64]. Training was determined to be vital for these interventions as it provided NPHWs the necessary skills, knowledge, and confidence to deliver health care for NCDs[61, 62]. Another necessary enabler was the availability of resources such as equipment to measure blood pressure or strips to check blood glucose, and a regular medicine supply[63]. The use of digital health tools demonstrated quality improvement and provided standardised and evidence-based care to communities[53]. The simplicity of the intervention, leveraging existing infrastructure and resources, and a collaborative care model facilitated the intervention's success[52,63]. Some studies provided free medicines or phones with prepaid data to support the implementation of the intervention. Nonetheless, the sustainability and scale-up of these interventions is debatable[52]. Support from senior managers and leaders was considered critical for the success of task-sharing interventions[63,65]. Legitimising the role of the NPHW to

ensure community acceptability, especially as they provide new services for conditions such as NCDs was considered essential[61].

Key barriers included the non-availability and erratic supply of medicines in public health facilities [53,60], distrust in the medicines available [61], a lack of equipment [63] and long waiting times for the PHC[64]. Furthermore, interventions that required space for delivery such as counselling or patient education found that many PHCs prioritised conditions such as HIV over NCDs and did not allocate any space within the health facility[60,62,65]. Some evaluations identified challenges such as poor management processes, poor relationships between PHC workers and conflict with higher level occupational groups[60,63]. NPHWs navigated this barrier with the support of community leaders or influential stakeholders[62]. Transportation for patients to attend the PHC, and health workers to visit patients at home, especially those who did not reside in the same community was identified as another barrier[63, 64]. Nurses also felt the need to improve mechanisms to store patient information at the PHC[63].

## Is task-sharing a cost-effective intervention?

Thirteen studies reported the costs involved [31,32,40,52,66–68] or evaluated the cost-effectiveness [28,69–73] of the intervention. Most (7/13) of these studies were conducted for interventions relating to mental health [40,68,72,74], three studies for hypertension [70,71,74], two for diabetes [69,73] and one for cardiovascular disease[66]. Economic assessment for the management of depression demonstrated that the intervention was cost-effective, with the task-sharing intervention costing US$120 less in the intervention arm than in the control arm in public health facilities[40]. A multi-country study on psychosis reported larger reductions in overall healthcare costs in the intervention group than in the control group. The study reported higher cumulative costs over the intervention period (US $627 per patient vs $526 in the control group[31]. The incremental cost-effectiveness ratio for task-shared care in Ethiopia indicated lower cost of –US$299·82 (95% CI –454·95 to –144·69) per unit increase in severe mental disorder clinical symptom severity (calculated using a brief psychiatric rating scale) from the health sector perspective at 12 months[28]. However, one study showed that the treatment was more costly per participant per year (US$117.16, 95%CI 94.05, 140.26) compared to enhanced usual care (US$85.30, 95%CI 55.98, 114.62; p = 0.04) and not cost-effective[68].

Task-sharing interventions for diabetes and hypertension indicated that involved trained NPHWs were cost-effective[69,71,73]. The incremental cost-effectiveness ratio for the CHW intervention for diabetes in American Samoa was calculated at $13,191 per quality-adjusted life year (QALY) gained, which is considered highly cost-effective compared to commonly accepted willingness-to-pay thresholds ranging from $39,000 to $154,353 per QALY in the study context. Some of the studies also measured the costs of intervention[69]. A study in South Asia estimated the cost of scale-up of a CHW task-sharing intervention for hypertension to be US$10.70, US$10.50, and US$4.70 per individual in Bangladesh, Pakistan, and Sri Lanka respectively[32]. Another study which trained CHWs to provide management for CVD estimated costs per individual at high-risk of CVD for three different models of care at 11 USD (CHW salary, training and physical measurement of CVD risk), 12 USD (basic model and medicines for CVD) and 14 USD (basic model, medicines and physician time)[66].

## Unintended consequences of task-sharing

As NPHWs were trained to take on new roles, in some contexts, this generated conflict with other staff[60,75]. For instance, studies in South Africa [60] and India [61] demonstrated that nurses were unhappy with the CHWs likely because their role was indirectly challenged by

that of CHWs who were trained to perform tasks similar to theirs. Furthermore, some studies reported challenges relating to insufficient remuneration of the CHWs, especially as they took on new roles[75]. A study which evaluated the management of hypertension at the PHC level where services for chronic disease including HIV are provided, demonstrated that vertically funded programs such as HIV and the poor standards of equipment in clinics compromised the quality of services provided by nurses[76].

Training improved the confidence and communication style and skills of CHWs, though some CHWs offered unsolicited information to patients[65]. Using digital health tools and sharing tasks with the PHC doctor for a common goal to improve health outcomes-built legitimacy for the CHW's new role[61].

### Risk of Bias of individual studies

Overall, over half (57%, 85/149) of studies had a low risk of bias, 8% (12/149) had a high risk of bias, and 35% (52/149) had some concerns with bias. There was a low risk of bias associated with randomisation (115/149), deviations (120/149), missing outcome data (129/149), measurement of outcomes (133/149), and selection of report result (131/149) in most studies. For full reporting of ROB results, please refer to S1 Appendix.

## Discussion

Our systematic review included studies that utilised NPHWs to prevent or control NCDs, and explored the barriers, facilitators and unexpected consequences of task-sharing. Our search identified 149 RCTs across 31 countries, of which 81% reported a positive primary outcome, demonstrating that task-sharing is an effective intervention for NCDs. NPHWs included CHWs, nurses, dietitians, nutritionists, and traditional faith healers. A sub-set of these studies which included economic analyses found that task-sharing can reduce the total costs of healthcare of patients with depression, anxiety, hypertension and diabetes and improve health outcomes in public facilities[28,69–72]. One study showed that task-sharing interventions were more costly than usual care [31], owing to the training and equipment required to upskill the workforce for providing quality health services.

Previous reviews on task-sharing have identified that it is effective for screening, prevention, and in some cases, the management of mental health conditions [77], hypertension [78], CVD [79], diabetes [80], cholesterol [81], cervical cancer [82] and other NCDs [11,83] - though effectiveness was not demonstrated for cholesterol-lowering interventions[81]. Task sharing has been achieved either by organising the available health workforce by expanding their current roles to include management of NCDs [53,54,74] or by employing additional resources such as community volunteers [84] or faith healers[31]. These models of care usually employ a multidisciplinary team of CHWs and nurses with or without physicians. Although our review found that 81% of the studies reported positive primary outcomes, indicating that task-sharing is an effective intervention for NCDs, a few studies reported neutral, mixed, and negative (one study) results. Various contextual factors at different levels seem to have contributed to these mixed outcomes. At the health system level, factors such as health infrastructure, the capabilities of NPHWs in implementing interventions, and human resource interventions (e.g., supervision and training) may affect effectiveness. Additionally, patient-level factors, such as engagement with interventions and adherence to treatment, also play a role in shaping the outcomes of task-sharing [48,54,59].

Furthermore, many of the task-sharing interventions were multifaceted, some aided by digital health to provide clinical decision support to the workforce[49,51,54,59], and others by the use of phone calls for health education, follow-up and medication adherence [55,85,86] to

improve health outcomes. Some studies focussed on a single disease or risk factor [38,87,88] while others evaluated task-sharing for a range of conditions[46,53,89–93]. Use of technology, training, and supervision of the health workforce were identified as facilitators.

However, the effectiveness of NPHWs in task-sharing interventions for NCDs can be influenced by whether they are dedicated solely to a given intervention or tasked with multiple duties within the broader health system. Studies where NPHWs received focused training and were assigned well-defined roles generally report positive outcomes, as these workers can concentrate on NCD-related tasks without competing responsibilities [54]. While digital health-related interventions improved access to effective health care and improved patient outcomes in most studies, scaling up the intervention would require considerable planning and funding to avoid 'pilotitis'[94]. Appreciation of the context and system-related issues such as non-availability of medicines or the need for a doctor to initiate treatment, as well as data integration with the sub-national or national health information system are important considerations.

As acknowledged by other reviews[11,83], a key takeaway was the macro-level and systemic barriers such as poor medicine supply, lack of equipment and infrastructure which impeded task-sharing. These issues directly influenced intervention outcomes, with some studies reporting disruptions in medication or equipment that hindered NPHWs' ability to deliver care [11,83]. Such barriers highlight the need to address systemic challenges, including supply chain inefficiencies. Ensuring a consistent supply of essential drugs, establishing an efficient distribution system, and providing training on proper use are critical steps to enhance the effectiveness of task-sharing interventions, particularly in resource-constrained settings [11].

Additional factors included lack of trust in the 'free medicines' provided by the public health system[61], low priority given to NCDs compared to communicable diseases [60,62] and additional costs involved in home visits for follow-up[63]. In fact, a recent assessment of Ethiopia's Health Extension Program services showed that better HIV program performance by CHWs was associated with lower uptake of NCD preventive services[95]. This finding supports the opinion that integration of new programs to existing service packages may spread resources too thinly[96]. This may jeopardise the success of existing health services resulting in worse health outcomes[95].

Having strong community engagement was found to circumvent some of these barriers[62]. As these multifaceted, complex, task-sharing interventions intrinsically depend on the interpersonal relationships of the healthcare teams, some studies found that if the roles of various team members were not clearly defined, it led to role conflict[60,75]. Other researchers have reported that the non-availability of protocols, lack of job description, differential financial incentives and the display of occupational superiority leads to role conflicts among the non-physician PHC team members[97].

Task-sharing is a well-accepted and an effective model of care which can help address the challenges of workforce shortages and inequities in healthcare access. The model has been embedded in the health system of several LMICs for decades to deliver care for communicable diseases and maternal and child health[98]. However, as communicable diseases require short-term care, adapting this model to address the long-term care needs of individuals with NCDs is essential. Decentralising services through task-sharing enables NPHWs to provide vital care in community settings, improving accessibility and continuity of care—key factors for managing chronic conditions effectively.

## Recommendations from this review

S3 Appendix documents detailed evidence about the tasks shared by each category of NPHWs for each type of NCD across the continuum of care (prevention, diagnosis, management, and rehabilitation) of patients with NCDs. Using a systems lens with a focus on task-sharing for

NCDs in LMICs, we have the following recommendations. At the **macro level**, national health policies need to include a specific policy for NCD related prevention, promotion, management, rehabilitation, and palliation. In order to implement these policies, countries need to invest in NCDs and allocate sufficient funds. As these NPHWs are also tasked with delivering additional services other than NCDs, including those related to communicable diseases, there is a need for health systems to focus on effective integration of services and systems.

At the **meso level**, implementors should ideally move from small scale pilots and trials to scaling up evidence-based solutions such as WHO Best buys through PHC as the platform for NCD care. Evidence demonstrates that digital health tools assist the health workforce to provide quality and standardised care and legitimises the role of CHWs. Availability of equipment, regular medicine supply and adequate space are necessary to build workforce and community trust in the health system. Furthermore, to motivate and retain the workforce, they need to be adequately remunerated.

At the **micro level**, our review highlights that all occupational groups need to have clear job descriptions with appropriate training and retraining of health workforce, especially NPHWs (e.g. CHWs, nurses) to improve their confidence, knowledge and skill set for basic NCD management at the community level. Accountability and community engagement were found to facilitate services. As team-based care requires close interaction and trust between team members, it is essential to provide guidance about how services will be integrated and how each occupation will function. The capability, opportunity, and motivation (COM-B) theory uses three interrelated domains which are linked to behaviour change. Capability includes physical and psychological capacity to engage in or perform an activity, motivation refers to automated and reflective brain processes that energize and direct behaviour, and opportunity refers to all the factors that lie outside of the control of an individual that influence change [75].

## Strengths and limitations

Our review includes all randomised controlled trials on task-sharing for NCD management reported in peer reviewed English, Spanish and French language journals. The strength of our report lies in its comprehensive scope as it is a large review encompassing a range of NPHWs and publications reporting both positive and neutral primary outcomes. It explored the barriers, facilitators, and unexpected consequences of task-sharing to NPHWs in the prevention and control of NCDs. Additionally, the overall risk of bias was low in majority of the studies included.

However, the study is not without limitations, most of which stem from the broader scope of the review, which may have missed specific details and nuances. One is the exclusion of studies published in languages other than English, Spanish, and French, which may have led to missing relevant examples from larger nations where research is conducted in other languages. Another limitation is related to combining all countries under the broad label of "LMICs," which overlooks critical differences in health systems, economic conditions, and cultural and social environments. A more detailed analysis of how task-sharing varies across specific regions and contexts would further enrich the findings. Additionally, very few RCTs reported process evaluation data, which is essential to understand the contextual factors and fidelity of the intervention. Details about training or retraining of the workforce, their supervision and remuneration was not discussed in the studies. Moreover, the diverse reporting in the included studies meant that we could not report disaggregated outcomes contributed by specific NPHWs. The other limitation of this review is that all the findings are based within a research context, which is usually challenging to scale-up due to a number of reasons including cost, adequate monitoring, health inequities, challenges in implementation due to shortage of workforce[99]. Additionally, the exclusion of implementation research may have led to the

omission of important contextual and real-world evidence that could provide valuable insights into the practical application of task-sharing interventions. Furthermore, few studies reported cost-effectiveness data which is important for assessing budget impact and the feasibility of scaling sustainably. Nonetheless, the study successfully attained its objectives as it focused on the task-sharing practices of NPHWs with different levels of training [9].

## Conclusions

Our review demonstrates that using task-sharing models of care involving trained NPHWs is effective and cost-effective in LMICs where NCDs are the leading causes of premature deaths and disability. Ultimately, task-sharing, should not be viewed as a task-dumping exercise to the 'lowest' occupational group, on the contrary, it ought to be designed and implemented as a team-based approach where all members are motivated, trained, remunerated and have the government and community's support to deliver their roles to the best of their ability.

## Supporting information

**S1 Appendix** – Search Terms.
(PDF)

**S2 Appendix** – Risk of bias results.
(PDF)

**S3 Appendix** – Evidence of effective task-sharing from this review.
(PDF)

## Acknowledgement

The Authors would like to acknowledge Giorgio Cometto and Alarcos Cieza for reviewing and commenting on this manuscript.

## Author contributions

**Conceptualization:** Rohina Joshi.

**Data curation:** Azeb Gebresilassie Tesema, Sikhumbuzo A. Mabunda, Kanika Chaudhri, Anthony Sunjaya, Samuel Thio, Kenneth Yakubu, Ragavi Jeyakumar, Myron Godinho, Renu John, Rohina Joshi.

**Formal analysis:** Azeb Gebresilassie Tesema, Sikhumbuzo A. Mabunda, Kanika Chaudhri, Anthony Sunjaya, Rohina Joshi.

**Funding acquisition:** Rohina Joshi.

**Project administration:** Rohina Joshi.

**Supervision:** Rohina Joshi.

**Validation:** Mai Eltigany, Martyna Hogendorf, Rohina Joshi.

**Visualization:** Azeb Gebresilassie Tesema, Sikhumbuzo A. Mabunda, Kanika Chaudhri, Anthony Sunjaya.

**Writing – original draft:** Kanika Chaudhri, Anthony Sunjaya, Rohina Joshi.

**Writing – review & editing:** Azeb Gebresilassie Tesema, Sikhumbuzo A. Mabunda, Samuel Thio, Kenneth Yakubu, Ragavi Jeyakumar, Myron Godinho, Renu John, Mai Eltigany, Martyna Hogendorf, Rohina Joshi.

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
