## [Decision Letter · Decision Letter 0]

4 Oct 2024

PGPH-D-24-02123

Task-sharing for non-communicable disease prevention and control in low- and middle-income countries in the context of health worker shortages: A systematic review

Dear Dr. Tesema,

Thank you for submitting your manuscript to PLOS Global Public Health. After careful consideration, we feel that it has merit but does not fully meet PLOS Global Public Health’s publication criteria as it currently stands. Therefore, we invite you to submit a revised version of the manuscript that addresses the points raised during the review process.

The reviewers have suggested providing more clarity on methods and presentation of results and nuances in the discussion. Please ensure these are addressed in the revised manuscript. 

We look forward to receiving your revised manuscript.

Kind regards,

Roopa Shivashankar, MD, MSc

Academic Editor

Journal Requirements:

**Please only choose the relevant sentences from below**

1. Please clarify all sources of funding (financial or material support) for your study. List the grants (with grant number) or organizations (with url) that supported your study, including funding received from your institution. 

2. State the initials, alongside each funding source, of each author to receive each grant.

3. State what role the funders took in the study. If the funders had no role in your study, please state: “The funders had no role in study design, data collection and analysis, decision to publish, or preparation of the manuscript.”

4. If any authors received a salary from any of your funders, please state which authors and which funders.

2. We note that your Data Availability Statement is currently as follows: "Data collected for the study is provided within the manuscript. Additional related documents are available in the supplementary section."

3. Please provide separate figure files in .tif or .eps format.

4. We have noticed that you have uploaded Supporting Information files, but you have not included a list of legends. Please add a full list of legends for your Supporting Information files after the references list. 

5. As required by our policy on Data Availability, please ensure your manuscript or supplementary information includes the following: 

6. Figure 2: please (a) provide a direct link to the base layer of the map (i.e., the country or region border shape) and ensure this is also included in the figure legend; and (b) provide a link to the terms of use / license information for the base layer image or shapefile. We cannot publish proprietary or copyrighted maps (e.g. Google Maps, Mapquest) and the terms of use for your map base layer must be compatible with our CC-BY 4.0 license. 

Additional Editor Comments (if provided):

Reviewers' comments:

Reviewer's Responses to Questions

**Comments to the Author**

1. Does this manuscript meet PLOS Global Public Health’s publication criteria ? Is the manuscript technically sound, and do the data support the conclusions? The manuscript must describe methodologically and ethically rigorous research with conclusions that are appropriately drawn based on the data presented.

Reviewer #1: Yes

Reviewer #2: Yes

2. Has the statistical analysis been performed appropriately and rigorously?

Reviewer #1: N/A

Reviewer #2: N/A

3. Have the authors made all data underlying the findings in their manuscript fully available (please refer to the Data Availability Statement at the start of the manuscript PDF file)?

Reviewer #1: Yes

Reviewer #2: Yes

4. Is the manuscript presented in an intelligible fashion and written in standard English?

Reviewer #1: Yes

Reviewer #2: Yes

5. Review Comments to the Author

Reviewer #1: This systematic review addresses an important issue in low- and middle-income countries (LMICs): the role of non-physician health workers (NPHWs) in delivering services related to non-communicable diseases (NCDs). The study provides valuable insights into the effectiveness of task-sharing for NCD prevention and control across 31 countries and various health conditions. However, several areas require further elaboration and refinement for a comprehensive understanding.

1. Research Question and Scope:

• The research question is relevant and addresses an essential gap in NCD service delivery. The inclusion of a wide range of diseases and multiple types of NPHWs provides a broad understanding of task-sharing's potential.

• The study combines findings under the umbrella of "LMICs," which could obscure critical differences in health systems, geography, and culture. A more granular analysis of how task-sharing varies across specific regions or health systems would enhance the generalizability of the findings. This should be addressed as a limitation.

2. Literature Search Strategy:

• The study's search strategy is comprehensive, covering multiple databases and grey literature.

• Under section of “Search Strategy”, participants, interventions, comparisons and outcomes need to be reorganized and presented as it is done in the protocol published on PROSPERO

3. Study Selection and Quality Assessment:

• A large number of studies were included, and the use of the Cochrane Risk of Bias Tool v2 is appropriate for assessing study quality.

• Review is limited to RCTs , cluster RCTs and their process evaluation. However, there is potential for selection bias considering the fact that many programs involving task sharing of NPHWs might have documented it as implementation research or health system research wherein trial designing was complex. Moreover, objectives of the review were qualitative in nature and this review doesn’t culminate in to pooling of an outcome by statistical means, so reason for exclusion of these studies is not clear. This should be addressed in the limitations section.

• Operational definition of task sharing used for selection of the study should be made more explicit.

• It seems criteria of sample size of 50 pertains to participants of the study, it should be made more explicit. Was there any criteria for number of NPHWs/CHWs involved in the study?

4. Data Extraction and Synthesis:

• The review provides a clear breakdown of the types of NPHWs involved (nurses, community health workers) and the conditions they managed (cardiovascular diseases, mental health, diabetes, etc.).

• However, putting all CHW as a single category may not be appropriate considering the heterogeneity in their selection criteria, nature of employment, job responsibilities, payment structure, type of health systems etc. Feasibility of subcategorization of CHW may be explored.

5. Results Presentation:

• The inclusion of cost-effectiveness studies is a valuable addition, demonstrating the economic benefits of task-sharing. The review reports that task-sharing reduced healthcare costs, which is critical for scaling interventions in LMICs. However for some studies only costs of intervention are mentioned. Also health economic outcome reporting should be similar across studies.

• The studies should not be arranged alphabetically but rather categorized by disease and activity in order to improve readability and comparison. (Table-1 and Table-2).

• Ensure consistency in summarizing outcomes in Table-2. The summary should be similar for the given outcome. e.g. for blood pressure outcome, for some studies only reduction is mentioned not supported by numbers, for some studies only mean reduction is mentioned while for some studies mean reduction along with 95% confidence interval is mentioned.

• A graphical illustration showing care pathway of selected NCDs and mapping of tasks which NPHWs and corresponding number of studies will improve understanding of the reader. This information in detail is given in the appendix which needs simplification in the form of graph as suggested.

• Typographical error in number of studies included. In abstract 427 is shown, it has to be 426 (399 +27).

Discussion

• The review needs a detailed analysis of mixed, neutral, or negative outcomes. A more nuanced discussion of these findings is necessary to understand the contextual factors influencing task-sharing's success or failure. This would also inform how interventions can be optimized for diverse settings, particularly in fragile health systems where mixed outcomes are more likely.

• More content to discussion needs to be added with regards to research context of studies, which may have recruited dedicated NPHWs with focus on the tasks of the study vis-à-vis NPHWs from the health system who have another competing tasks to accomplish.

Reviewer #2: The manuscript provides a valuable contribution to understanding task-sharing for non-communicable disease management in low- and middle-income countries, addressing a crucial issue given the global shortage of healthcare workers. The review offers timely insights into how non-physician health workers can help bridge healthcare gaps, making it relevant for policymakers and practitioners.

While the manuscript is strong, there are areas where clarity and impact could be improved. Though meta-analysis was not in scope of this study, enhancing the background with additional context on the need for task-sharing would strengthen the rationale. A more in-depth analysis of the data and results, supported by visual aids, would make the findings clearer and more engaging for readers. Additionally, expanding the discussion around the challenges and barriers faced during task-sharing interventions would provide a more balanced perspective.

Overall, the manuscript makes an important contribution to the field, and with some revisions, it has the potential to significantly influence policy and practice in global health.

6. PLOS authors have the option to publish the peer review history of their article (what does this mean? ). If published, this will include your full peer review and any attached files.

**Do you want your identity to be public for this peer review?** For information about this choice, including consent withdrawal, please see our Privacy Policy .

Reviewer #1: No

Reviewer #2: **Yes: ** Ashish Krishna

---

## [Decision Letter · Decision Letter 1]

28 Jan 2025

Task-sharing for non-communicable disease prevention and control in low- and middle-income countries in the context of health worker shortages: A systematic review

PGPH-D-24-02123R1

Dear Deepavali A Dhond,

We are pleased to inform you that your manuscript 'Task-sharing for non-communicable disease prevention and control in low- and middle-income countries in the context of health worker shortages: A systematic review' has been provisionally accepted for publication in PLOS Global Public Health.

Best regards,

Roopa Shivashankar, MD, MSc

Academic Editor

Reviewer Comments (if any, and for reference):

Reviewer's Responses to Questions

**Comments to the Author**

1. If the authors have adequately addressed your comments raised in a previous round of review and you feel that this manuscript is now acceptable for publication, you may indicate that here to bypass the “Comments to the Author” section, enter your conflict of interest statement in the “Confidential to Editor” section, and submit your "Accept" recommendation.

Reviewer #1: All comments have been addressed

Reviewer #2: All comments have been addressed

2. Does this manuscript meet PLOS Global Public Health’s publication criteria ? Is the manuscript technically sound, and do the data support the conclusions? The manuscript must describe methodologically and ethically rigorous research with conclusions that are appropriately drawn based on the data presented.

Reviewer #1: Yes

Reviewer #2: Yes

3. Has the statistical analysis been performed appropriately and rigorously?

Reviewer #1: N/A

Reviewer #2: Yes

4. Have the authors made all data underlying the findings in their manuscript fully available (please refer to the Data Availability Statement at the start of the manuscript PDF file)?

Reviewer #1: Yes

Reviewer #2: Yes

5. Is the manuscript presented in an intelligible fashion and written in standard English?

Reviewer #1: Yes

Reviewer #2: Yes

6. Review Comments to the Author

Reviewer #1: The authors have provided detailed response to the comments and also revised the relevant portions in the manuscript.

Reviewer #2: (No Response)

7. PLOS authors have the option to publish the peer review history of their article (what does this mean? ). If published, this will include your full peer review and any attached files.

**Do you want your identity to be public for this peer review?** For information about this choice, including consent withdrawal, please see our Privacy Policy .

Reviewer #1: No

Reviewer #2: **Yes: ** Ashish Krishna
